# Immuno-metabolic dendritic cell vaccine signatures associate with overall survival in vaccinated melanoma patients

Juraj Adamik[1], Paul V. Munson[1], Deena M. Maurer[1], Felix J. Hartmann [ID][2], Sean C. Bendall [ID][3], Rafael J. Argüello [ID][4] & Lisa H. Butterfield [ID][1,5] ✉

Efficacy of cancer vaccines remains low and mechanistic understanding of antigen presenting cell function in cancer may improve vaccine design and outcomes. Here, we analyze the transcriptomic and immune-metabolic profiles of Dendritic Cells (DCs) from 35 subjects enrolled in a trial of DC vaccines in late-stage melanoma (NCT01622933). Multiple platforms identify metabolism as an important biomarker of DC function and patient overall survival (OS). We demonstrate multiple immune and metabolic gene expression pathway alterations, a functional decrease in OCR/OXPHOS and increase in ECAR/glycolysis in patient vaccines. To dissect molecular mechanisms, we utilize single cell SCENITH functional profiling and show patient clinical outcomes (OS) correlate with DC metabolic profile, and that metabolism is linked to immune phenotype. With single cell metabolic regulome profiling, we show that MCT1 (monocarboxylate transporter-1), a lactate transporter, is increased in patient DCs, as is glucose uptake and lactate secretion. Importantly, pre-vaccination circulating myeloid cells in patients used as precursors for DC vaccine generation are significantly skewed metabolically as are several DC subsets. Together, we demonstrate that the metabolic profile of DC is tightly associated with the immunostimulatory potential of DC vaccines from cancer patients. We link phenotypic and functional metabolic changes to immune signatures that correspond to suppressed DC differentiation.

Dendritic cells (DC) are central to innate and adaptive immunity through recognition of pathogen and danger-associated signals for orchestrating inflammatory responses and priming antigen-specific T cells[1]. Cancer vaccines are designed to promote long-lived anti-tumor-specific T-cell responses, but despite being safe, these vaccines generally lack durable clinical efficacy[2]. Therefore, a better mechanistic understanding of DC-based vaccine generation and development of biomarkers for immune as well as metabolic monitoring of the patient-derived precursors and DC is critical[3]. In addition, melanoma remains a serious health risk with a continued increase in incidence for the past 30 years[4], and while checkpoint blockade revolutionized treatment, a significant proportion of patients do not respond and/or acquire resistance to checkpoint therapies[5].

DC maturation results in upregulation of major histocompatibility complexes (MHC), costimulatory receptors (CD86, CD80, CD40, ICOSL) and secretion of cytokines[6]. This highly coordinated process

[1]Parker Institute for Cancer Immunotherapy, San Francisco, CA 94129, USA. [2]Systems Immunology and Single-Cell Biology, German Cancer Research Center (DKFZ), Heidelberg, Germany. [3]Department of Pathology, Stanford University, Palo Alto, CA 94304, USA. [4]Aix Marseille Univ, CNRS, INSERM, CIML, Centre d'Immunologie de Marseille-Luminy, Marseille, France. [5]Department of Microbiology and Immunology, University of California San Francisco, San Francisco, CA, USA. ✉e-mail: lisa.butterfield@ucsf.edu

requires metabolic adaptations to meet the energy demands associated with phenotypic and morphologic changes that enable DC functional specialization for mounting immune responses. Several studies demonstrate that modulation of cellular metabolic programs is required for energy demands associated with functional changes in transcriptional and biosynthetic pathways in DC for survival and effective T-cell priming capacity[7].

A major shift from oxidative phosphorylation (OXPHOS) to aerobic glycolysis was shown to be required upon Toll-like receptor (TLR) activation and antigen presentation in murine bone marrow-derived DCs (mBMDCs)[8-13]. While glycolytic metabolism is a hallmark of mBMDC activation, this phenomenon does not directly translate to human DC[14-17] and alterations in metabolic wiring have been attributed to distinct inflammatory and tolerogenic states as well as myeloid/DC subtypes and species-specific differences[8,11,15,16,18-20]. Furthermore, diverse metabolic programs and mitochondrial reprogramming underlie cellular fate and function of distinct DC subtypes[21]. Metabolic differences associated with deregulated OXPHOS, glycolysis and fatty acid oxidation (FAO) programs were also shown to influence anti-inflammatory phenotype of tolerogenic DCs (tolDC), which maintain immune tolerance by inhibiting effector and autoreactive T cells, and polarizing development of regulatory T-cell (Treg) responses[22].

Evidence for global reprograming of inflammatory DC activation stems primarily from transcriptional profiling and metabolic studies often rely metabolic respiration by means of metabolite tracing and/or oxygen consumption (OCR)/extracellular flux analyses (ECAR)[23-25].

While providing invaluable insights to the field of immunometabolism, the technical limitations of bulk cellular measurements are not able to adequately capture the newly-appreciated phenotypic and functional diversity associated with the heterogeneous nature of in vitro DC culture systems[25,26]. Emergence of single-cell approaches using RNA sequencing and high-dimensional mass cytometry by time of flight (CyTOF), and fluorescent cytometry-based techniques enables robust estimation of immuno-metabolic states of individual cells in the context of heterogeneous cell populations[17,27-31]. We recently coupled novel single-cell energetic metabolism by profiling translation inhibition (SCENITH)[28] and CyTOF-based single-cell metabolic regulome profiling (scMEP)[30] to integrate functional measurements to quantify metabolite transporters and enzymes across major cellular metabolic axes in both human inflammatory and tolerogenic DC[17]. We identified coordinated activation of multiple metabolic pathways along distinct stages of monocytic DC differentiation and maturation. Our mapping of functional metabolic states and the underlying metabolic protein regulome showed that elevated phospho-mTOR:AMPK ratio with upregulation of OXPHOS, glycolytic and fatty acid oxidation metabolism underlies the metabolic hyperactivity of the immunosuppressive phenotype of tolerogenic DC[17].

As key regulators of immune homeostasis, monocytic DC have been critical resources for diverse cell therapy applications including priming antitumor T-cell responses as cancer vaccines[32], or in the opposing role as tolerogenic cells, promoting immune suppression for organ transplantation and autoimmune disease treatment[33]. We now utilize both bulk and single cell metabolic profiling of melanoma patient DC and identify metabolic skewing and increased glycolysis which impacts overall survival in melanoma patients receiving ex vivo DC vaccines. We also determine the baseline metabolic state of circulating monocyte and DC subsets in these patients and healthy donors identify similar metabolic dysfunction. These data suggest that the cancer state induces skewed myeloid cell metabolism, and that ex vivo culture and maturation of such monocytes to DC vaccines by current approaches may not fully reconstitute the optimal balanced cellular metabolic activity, nor the immune stimulatory phenotype of DC.

## Results

### HD and melanoma patient mDC exhibit significant differences in global transcriptional profiles

We recently performed a clinical trial testing melanoma antigen engineered DC vaccines[34]. These DC vaccines were cultured for 7 days ex vivo from circulating myeloid cells before antigen loading and injection. In depth transcriptomic and immune-metabolic profiling was applied to analyze maturation states of melanoma patient-derived IFNγ + LPS matured DC (mDC) used for the autologous vaccine preparation. Figure 1A shows a schematic of the DC maturation protocol with time points used for the four profiling methods. Microarray profiling of melanoma patient mDC revealed differential gene expression of 2077 genes (Fig. 1B), which reflects the global phenotypic and transcriptomic changes during DC maturation[35-37]. Differential expression of 82 genes enriched in hypoxia-related pathways and biosynthetic processes was further detected in adenovirally engineered (post-maturation) DC (Supplementary Fig. 1B). We focused our analysis on mDC and not the adenovirally engineered DC because most transcriptional changes occurred with maturation, the data would be more broadly applicable, and to compare our melanoma data to the healthy donor (HD) dataset. Differences between healthy donors (3 day) and melanoma patient mDC (5 days) iDC cultures do not impact our results as studies examining differences in immature iDC generation revealed that monocytes cultured in the presence of IL-4 + GM-CSF within 48 h exhibit iDC characteristics and upon maturation these cells displayed a fully mature mDC immunophenotype[38]. Comparison of melanoma patient with the publicly available HD mDC microarray profiles[39] further revealed that 725 upregulated and 818 downregulated genes were specific to the melanoma mDC (Fig. 1B). gProfiler pathway enrichment analysis of HD mDC showed significant upregulation MHC class I antigen-receptor processing/presentation and CCR5 chemokine receptor binding pathways. In contrast, VEGFA, TGFβ receptor, NLRP3 inflammasome and Oncostatin M signaling were selectively upregulated, while antigen processing and pattern recognition receptor activity genes were downregulated in melanoma mDC (Supplementary Fig. 1A). Gene set enrichment analysis (GSEA) of overlapping signatures showed selective downregulation of metabolic genes involved in TCA cycle and electron transport chain (ETC)/Oxphos in HD and FA/phospholipid metabolism and PPAR pathway in melanoma mDC (Fig. 1C). In the antigen presenting cells, (APC)/Cytokine/Chemokine/Immune category, differences in IL-2, IL-3 and STAT3 signaling pathways in melanoma and Wnt, Rho GTPases MAPK4/6 signaling pathways in HD mDC were observed. In addition to HD vs melanoma differences, we explored differential gene signature correlations with clinical outcome groups ("good" (PR + SD > 6 mo.+ non-recurrent NED); "bad" (PD + SD ≤ 6 mo. + recurrent NED[34,40,41])). Among enriched immune and metabolic pathways, LPS/inflammatory response, NFκB targets, DC maturation, VEGF/Hypoxia, APC/MHC/Interleukin/Matrisome/Intergins and FAO/Sphingolipid metabolism associated with favorable clinical outcome (Fig. 1D). In contrast, genes in the DNA Repair, TCA/ETC, mRNA processing, Interferon signaling and Golgi-ER transport/Glycosylation category were upregulated in the worse outcome mDC. To evaluate gene signatures as potential biomarkers for separating clinical outcome groups, we identified 57 upregulated genes in the good outcome mDC which included cytokine activity (IL1A, CCL24, CXCL6, CXCL5, IFNG), and extracellular matrix disassembly (MMP1, MMP9, MMP110, MMP112) immunoregulatory genes (Supplementary Fig. 1C). Gene set variation analysis (GSVA) revealed that this gene signature was significantly upregulated in the good outcome mDC, but additional analysis will be required to further evaluate this gene set as a predictive signature of response for DC cancer vaccines.

While necessarily descriptive, these microarray differences indicated that many signaling pathways associated with cellular metabolism were important to examine functionally.

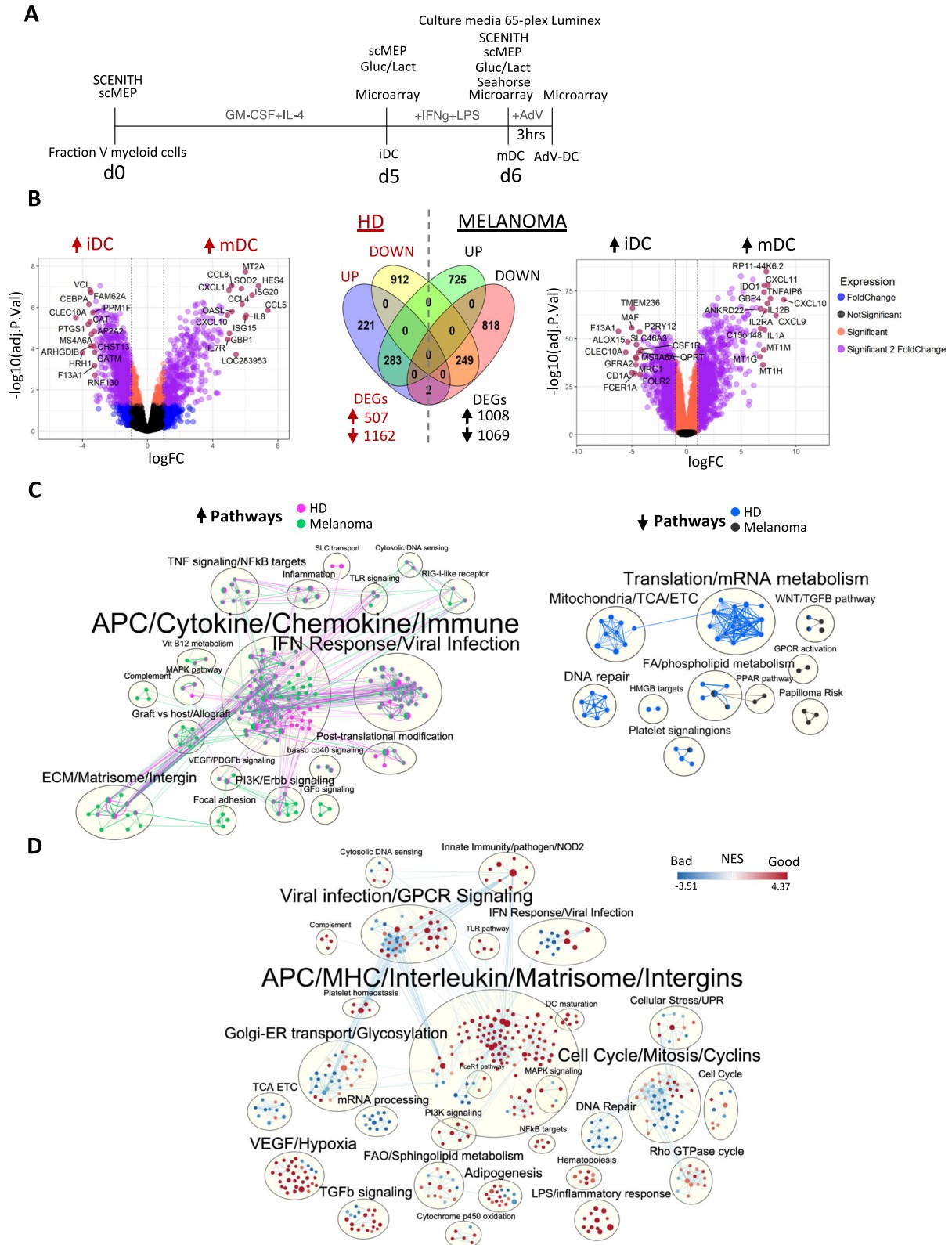

**Decreased mitochondrial metabolism and reduced FAO distinguish melanoma mDC from HD**

To confirm that the altered metabolic gene expression profile of melanoma patient mDC had downstream functional impact, we performed an assessment of mitochondrial and glycolytic metabolism in patient and HD cells using the Seahorse assay which measures mitochondrial respiration cellular oxygen consumption (OCR) and extracellular acidification (ECAR) to measure glycolysis. There were no differences in oxygen consumption rate (OCR)-derived parameters between melanoma and HD mDC, yet melanoma mDC demonstrated a trend towards decreased maximal oxygen consumption rate and spare respiratory capacity (Fig. 2A). A significant increase in basal glycolysis was observed in both melanoma patient clinical groups, while the glycolytic capacity was significantly increased primarily in bad outcome mDC (Fig. 2B).

**Fig. 1 | Gene expression profiling of mature DC from HD and melanoma patients. A** Conceptual overview of ex vivo mDC culture conditions with indicated time points used for profiling methods used in this study including microarray, Seahorse assay, culture supernatants Luminex assay, glucose and lactate measurements (Gluc/Lact), SCENITH and scMEP. **B** Volcano plots show differential gene expression at false discovery rate threshold of 5% by fold change (logFC) and adj.p-value (−log10(adj.p)) comparing changes between mDC vs iDC for HD (magenta/blue, n = 4) and melanoma (green/black, n = 35) patients. Dark gray dots denote non-significant genes, orange dots denote significant genes with fold change ≤2 and magenta dots indicate genes with fold change ≥2 and adj.p value ≤ 0.05.

Labeled are top 15 significant genes. **C** Summary of significantly upregulated and downregulated pathways (adj.p < 0.05) with overlapping gene sets identified by GSEA/MSigDB analysis between mDC vs iDC from HD (magenta/blue, n = 4) and melanoma (green/black, n = 35) patients. **D** Summary of significantly (adj.p < 0.05) different GSEA/MSigDB pathways between good (PR/SD/NED1, n = 13) and bad (PD/NED2, n = 17) outcome groups in mature mDC. The color-coding scale denotes magnitude of normalized enrichment score (NES) for each pathway with red and blue colors corresponding to enrichment in good and bad outcome mDC groups, respectively.

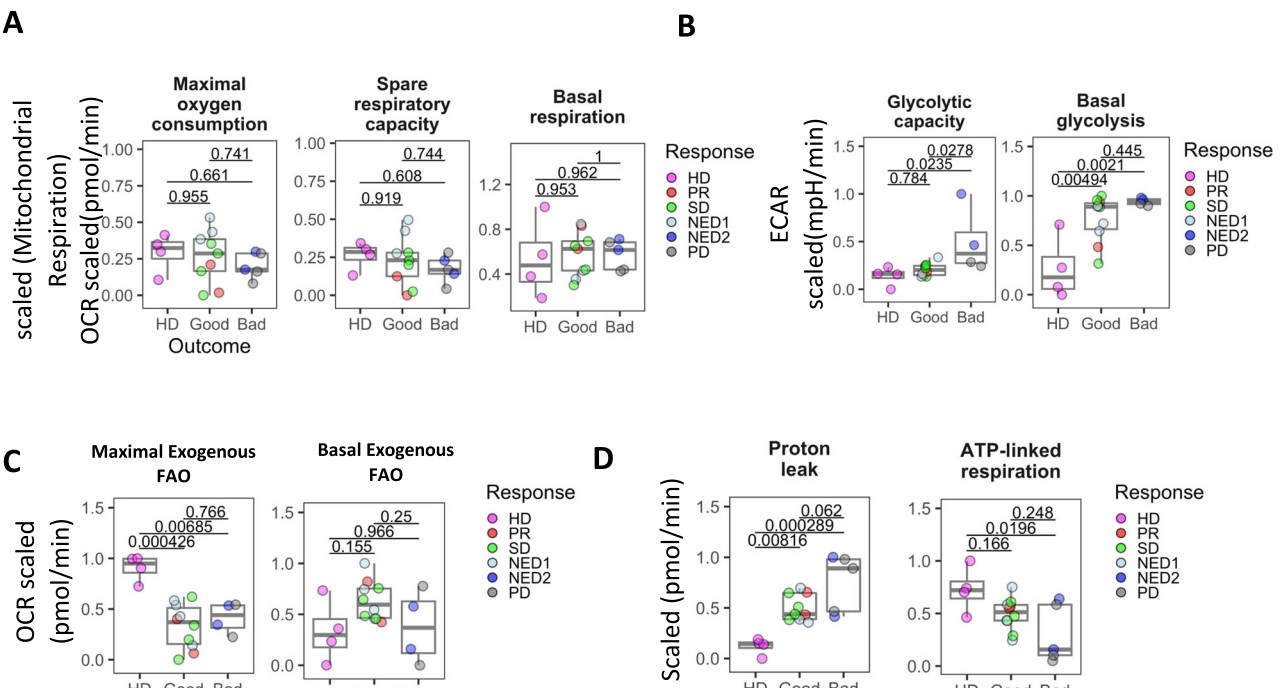

**Fig. 2 | Seahorse metabolic profiling HD and melanoma patient-derived mDC.** Box plots represent distribution of Seahorse metabolic measurements in mDCs stratified by clinical outcome. **A** Scaled values for oxygen consumption rate (OCR) parameters for maximal oxygen consumption rate, spare respiratory capacity and basal respiration, **B** glycolytic capacity and basal glycolysis derived from extracellular acidification rate (ECAR) measurements, **C** OCR-derived exogenous fatty acid oxidation and **D** proton leak and ATP-linked respiration changes are shown for mDCs from heathy donor (HD, n = 4), good (PR/SD/NED1, n = 9) and bad (PD/NED2, n = 6) outcome groups. Box plots indicate 1st, 2nd and 3rd quartile; whiskers indicate minimum and maximum. Multi-group comparisons were tested by one-way ANOVA with Tukey's *post-hoc* test. Source data are provided as a Source Data file.

To assess the capacity of mDC to oxidize exogenous fatty acids, we evaluated changes in OCR after addition of palmitate-BSA. Melanoma mDC exhibited significantly reduced ability to metabolize long-chain fatty acids compared to HD (Fig. 2C). Sequential addition of the ATP synthase inhibitor oligomycin enabled us to determine changes in proton leak, which was very low in HD, but significantly enhanced in a stepwise fashion in good and more so in bad outcome melanoma mDC (Fig. 2D). ATP-linked respiration exhibited significant decrease in bad outcome groups, which can indicate low ATP demand or damage to the ETC, which would prevent the flow of electrons and result in the observed decrease in OCR[42] (Fig. 2D). Together, the reduced FAO utilization activity, increased glycolytic capacity with the increased proton leak across the membrane and reduced ATP-linked respiration further supports mitochondrial bioenergetic dysfunction in melanoma derived mDC.

**Increased mitochondrial metabolism, FAO and glutaminolysis in mDC associate with increased survival in melanoma patients**
We employed the single-cell energetic metabolism by profiling translation inhibition (SCENITH™) assay to both further validate our Seahorse observations, and also determine the impact of metabolic

alterations on the immune phenotype of melanoma patient mDC[17,28,43]. The use of metabolic inhibitors 2DG, Oligomycin, Etomoxir and CB-839 in SCENITH enabled us to derive percentual parameters of metabolic activity in mDC. Consistent with our previous study, mitochondrial dependence was the highest metabolic process (80%) in mDC[17] (Supplementary Fig. 1A). We observed a close to significant decrease in mitochondrial dependence from 84.4% to 76.4%, with corresponding increase in glycolytic capacity from 15.6% to 23.6%, and 7% decrease in glutaminolysis dependence in bad outcome groups (Fig. 3A). Consistent with the Seahorse analysis, HD DC exhibited trends towards increased mitochondrial dependence, with reduced glycolytic capacity and significant increase in glutaminolysis dependence compared to melanoma mDC (Supplementary Fig. 1B). SCENITH metabolic parameters were divided into binary high and low categories based on selected optimal cutoff values using the maximally selected rank statistics[44] (Supplementary Fig. 2C). Cox's proportional-hazards models based on these binary categories show that higher mitochondrial dependence in patient mDC was significantly associated with longer OS and PFS rate. FAO and glutaminolysis dependence showed close to significant values (Fig. 3B). Kaplan–Meier (KM) survival analysis comparing SCENITH metabolic differences further confirmed significant

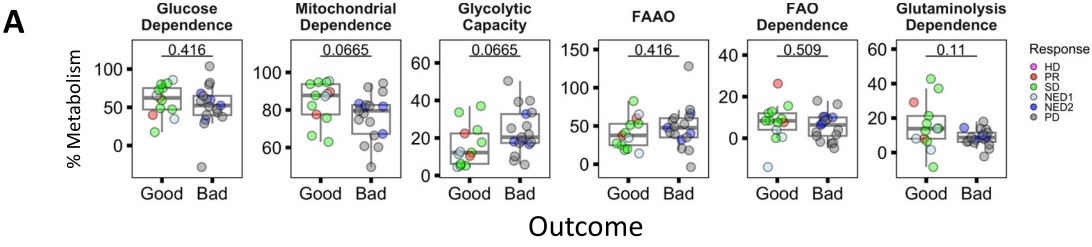

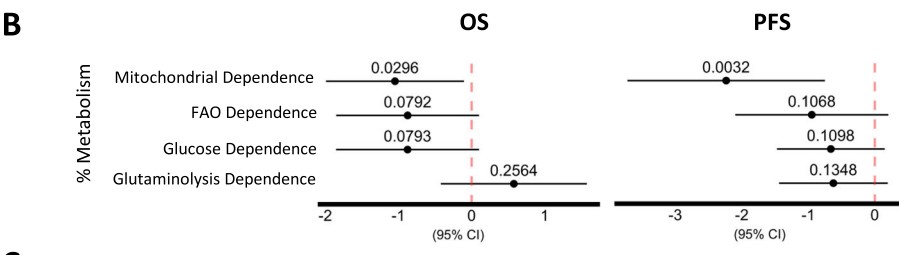

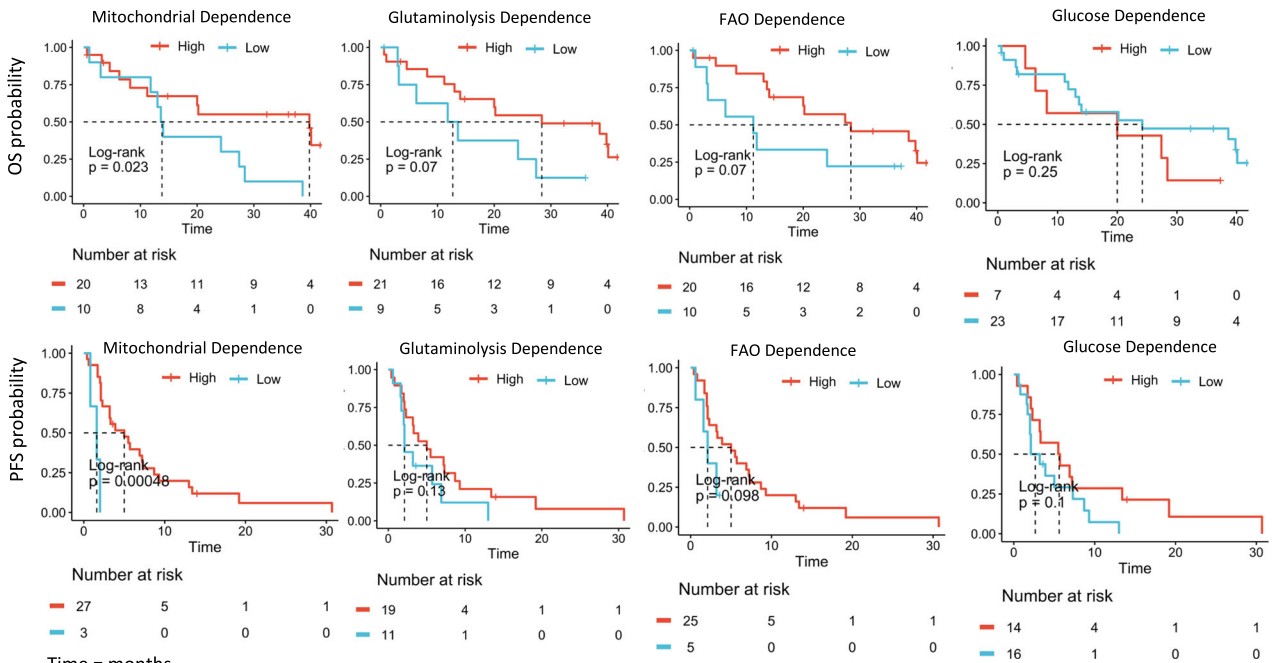

**Fig. 3 | SCENITH profiling and associations of metabolic profiles in mDC with OS and PFS in melanoma patients. A** Box plots represent median expression values for changes in percentual SCENITH parameters in mature mDC between good (PR/SD/NED1, n = 13) and bad (PD/NED2, n = 17) outcome groups. Statistical significance was tested using Two-tailed Student's *t*-test. Box plots indicate 1st, 2nd and 3rd quartile; whiskers indicate minimum and maximum. **B** Forest plot summarizing univariate Cox regression analyses for effects of high and low metabolic profiles

(SCENITH) in mDC on overall (OS) and progression free (PFS) survival with intercept points, *p*-values and 95% confidence intervals indicated (n = 30). **C** Kaplan–Meier survival analysis of OS and PFS comparing the survival benefits of metabolic profiles (SCENITH) in mDC (n = 30). log-rank test was used to compare statistical difference between the Kaplan–Meier curves. Source data are provided as a Source Data file.

associations between mitochondrial dependence (as well as trending FAO and glutaminolysis dependence) with longer OS and PFS rate (Fig. 3C). In analyzing the clinical trial results, ex vivo ELISPOT assays were performed to detect IFNγ-producing CD8 and CD4 T-cell responses specific to the DC vaccine loaded melanoma-associated antigens Tyrosinase, MART-1 and MAGE-A6. While we did not detect significant associations between metabolic parameters and melanoma antigen-specific T-cell responses, increased mitochondrial and FAO dependence showed a trend towards increased T-cell responses in CD8 and CD4 T cells respectively (Supplementary Fig. 3A).

## Melanoma mDCs with the highest glycolytic capacity exhibit aberrant expression of DC immune markers

SCENITH assay analysis integrated a full spectrum of DC phenotypic markers and the co-expression patterns of immune and signaling markers. The underlying changes in metabolic percentual parameters as well as clinical outcome and melanoma antigen-specific T-cell responses in melanoma compared to HD mDC were analyzed (Fig. 4A). Several immune and co-stimulatory molecules, including HLA-DR, CD86, CD206, CD40 as well as the inhibitory checkpoint molecule ILT3 and were significantly over-expressed in worse outcome patient

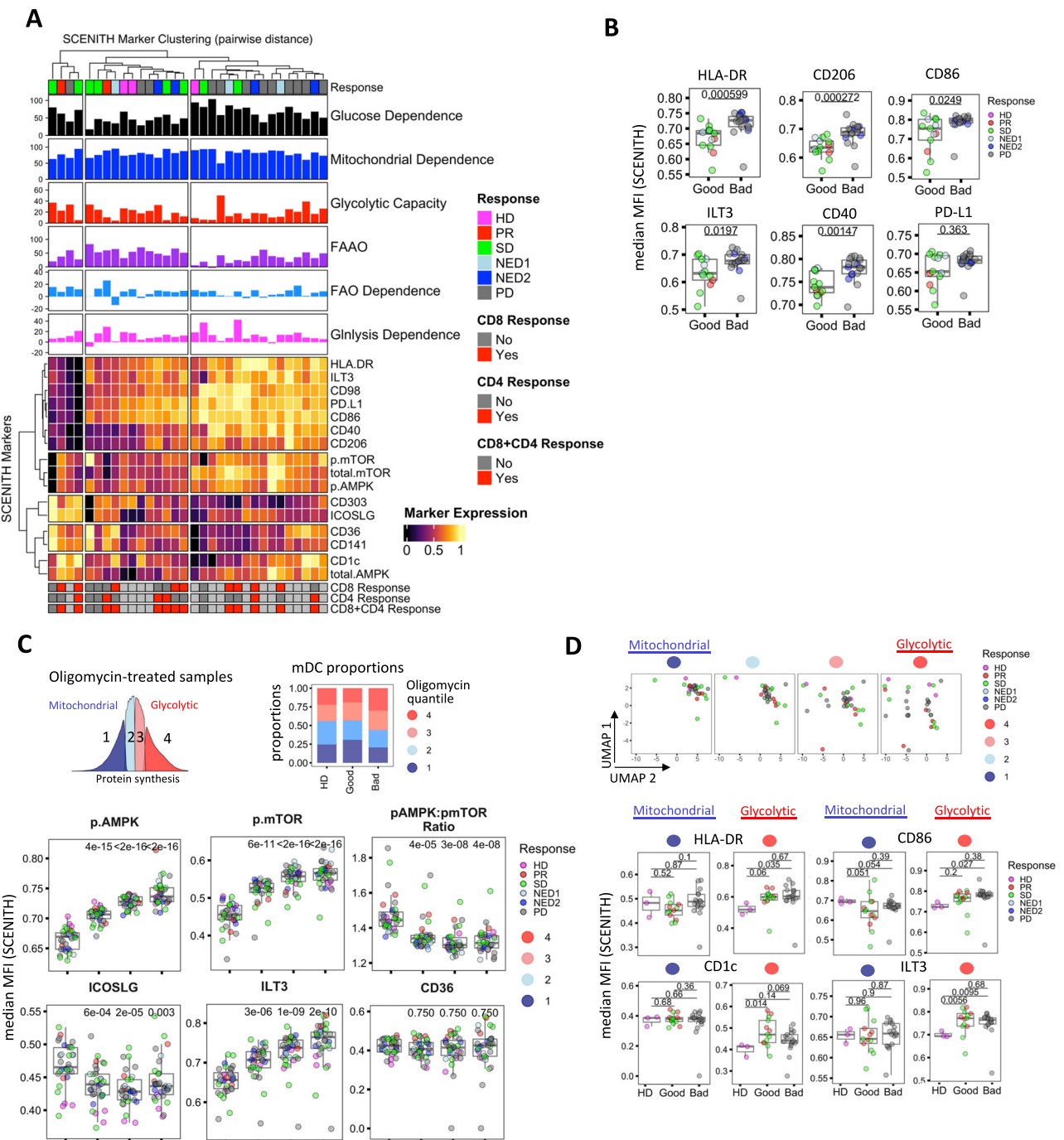

mDC (Fig. 4B). However, similar to the SCENTH metabolic parameters results, expression of these molecules did not exhibit significant association with melanoma antigen-specific T-cell responses (Supplementary Fig. 3B). Therefore, we analyzed the immune and metabolic expression profiles in more detail, at the level of specific mitochondrial vs glycolytic SCENITH profiles of mDC in oligomycin-treated samples (Fig. 4C). High glycolytic capacity is attributed to cells that are able to sustain high levels of protein synthesis after treatment with the mitochondrial inhibitor Oligomycin. In contrast, cells that block protein synthesis are "mitochondrial dependent" and not able to use or switch effectively to aerobic glycolysis (Fig. 4C)[28]. Analysis of mDC cell proportions showed that the worse responders contained the highest number of cells in the highest glycolytic quantile as compared to HD and good disease outcome groups (Fig. 4C).

To gain mechanistic insights into signaling pathways regulating OXPHOS and glycolytic melanoma mDC metabolism, we employed antibodies recognizing the total and phosphorylated forms of AMPK (Thr-183/172) and p-mTOR (Ser-2448) (Supplementary Table 1). These molecules were identified as a key regulatory node in our recent analysis of HD DC polarized to either inflammatory or tolerogenic profiles[17]. Phosphorylated AMPK and mTOR levels were gradually elevated in increasingly glycolytic cells, but the ratio of the two factors was skewed towards increased p-AMPK in cell populations with the highest mitochondrial dependence (Fig. 4C). This is consistent with increased mitochondrial dependence (Fig. 3A) in good outcome mDC, as well as our previous study that demonstrated an increased p-AMPK:p-mTOR ratio plays an important role in maintaining mitochondrial metabolism of differentiated mDC[17].

**Fig. 4 | SCENITH immune-metabolic profiling of glycolytic and mitochondrial-dependent mDC populations. A** Integrated clustering heatmap of median MFI expression for collection of SCENITH phenotyping markers (marker/antibody information is available in Supplementary Table 1) in mDC from heathy donor (HD, $n = 3$), good (PR/SD/NED1, $n = 13$) and bad (PD/NED2, $n = 17$) outcome groups. HD and patient response indications and absence (No) or presence (Yes) of patient-derived melanoma antigen (MA)-specific CD8, CD4, combined CD8 + CD4 IFN-γ T-cell responses, as defined in materials and methods, are annotated. SCENITH percentual metabolic profiles are represented as bar graphs on top of the heatmap. **B** Box plots represent differences in expression of median MFI expression profiles for immune-phenotyping in mDC between good (PR/SD/NED1, $n = 13$) and bad (PD/NED2, $n = 17$) clinical groups. Shapiro-Wilk test was used to assess data normality, Two-tailed Wilcoxon signed-rank test (non-normal data) and Two-tailed Student's t-test (normal data) was used for statistical analysis. **C** Protein synthesis histogram represents puromycin MFI profile for cultured mDC, which were treated with oligomycin. Protein synthesis profiles in oligomycin samples were binned into 4 quantiles, which represent metabolic states of mDC ranging from glycolytic (red population) to mitochondrial-dependent (blue) populations. Bar graphs represent proportions of cells within each oligomycin quantile within clinical response group. Box plots represent differences in expression of median MFI expression profiles for signaling and immune-phenotyping markers in HD and melanoma mDC among oligomycin quantiles. Each oligomycin quantile contains heathy donor (HD, $n = 3$), good (PR/SD/NED1, $n = 13$) and bad (PD/NED2, $n = 17$) samples. Annotation e-represents scientific annotation to the power of. **D** Immune marker-based uniform Manifold Approximation and Projection (UMAP) clustering of HD and melanoma mDC within each oligomycin quantile (indicated in Supplementary Table 1). Box plots represent differences in expression of median MFI expression profiles for immune markers in mDC between heathy donor (HD, n = 3) vs. good (PR/NED1/SD, $n = 13$) and bad (PD/NED2, $n = 17$) response groups in respective glycolytic and mitochondrial metabolic quantiles. In **B–D**, Box plots indicate 1st, 2nd and 3rd quartile; whiskers indicate minimum and maximum. Pairwise comparisons against a HD reference group in (**C**, **D**) were calculated using Two-tailed Student's t-test with Holm–Bonferroni correction. Source data are provided as a Source Data file.

Because p-AMPK is a well-established positive regulator of mitochondrial health[45] and metabolism[46–48], we employed Dorsomorphin to further analyze the consequences of p-AMPK inhibition on both immune and metabolic phenotype of mDC. Increased concentrations of Dorsomorphin inhibited p-AMPK phosphorylation, and resulted in reduced expression of several DC immune markers including HLA-DR, CD86, PD-L1 and CD206 in HD mDC (Supplementary Fig. 4A). p-AMPK inhibition also exhibited small but significant decrease in mitochondrial mass along with increase in glucose and decrease in lactate levels in media without impact on mDC viability (Supplementary Fig. 4A).

Dimensionality reduction of the four metabolic mDC states solely based on 12 immune DC surface markers showed that patient mDC in the glycolytic groups are more phenotypically diverse compared to the more uniform mitochondrial populations (which also clustered in the vicinity of the HD samples (Fig. 4D)). We compared immune marker expression among HD and clinical outcome groups in the highest glycolytic and mitochondrial populations. DC markers HLA-DR, CD86, CD1c, ILT3 and CD40 did not significantly differ among the outcome groups in mitochondrial populations, however their expression was significantly elevated in the melanoma bad and good outcome groups as compared to HD in the glycolytic cell states (Fig. 4C). As suggested by more uniform UMAP clustering, the mitochondrial patient mDC outcome groups exhibited less variation in the overall immune marker expression profiles and trended toward downregulation as compared to HD (Fig. 4D, Supplementary Fig. 4B). This single cell-based analysis approach provides further insight into the bulk Seahorse measurements and initial SCENITH results (Figs. 2A–C and 3A) to show the effects of underlying changes in glycolytic metabolism on the immune phenotypes of patient-derived mDC that would be otherwise be impossible to detect. While we and others have previously shown that p-AMPK associates with mitochondrial metabolism in maturing DC[48–50], we further showed that inhibition of p-AMPK resulted in impaired expression of surface immune phenotype and reduced mitochondrial mass of mDC.

Collectively, these results suggest that inhibition of mitochondrial metabolism results in impaired expression of surface DC immune phenotype, and that good outcome mDC exhibit higher proportion of mitochondrial cells in cultures as compared to bad outcome mDC from patients. The mitochondrial mDC cluster more uniformly with HD groups as compared to the glycolytic cell populations, which represent less uniform cell populations and exhibit variable expression of multiple immune surface markers.

## Distinctions between HD and melanoma DC are reflected by changes in the metabolic regulome

In parallel with SCENITH, we employed mass cytometry-based single-cell profiling of the metabolic regulome to integrate functional metabolic changes with quantification of metabolite transporters, enzymes and signaling factors across major cellular metabolic axes in immature and mature DC states[17,30] (Supplementary Table 2, Fig. 5A). Heatmap clustering using solely metabolic molecules enabled us to visualize patient iDC and mDC-specific scMEP regulome differences with overlayed immune phenotypes. While we did not observe a clinical outcome specific clustering trend, HD mDC cells grouped together along with several good outcome patients. We noted that in the mDC, scMEP markers segregated cell populations with higher HLA-DR vs CD11b and CD14 expression profiles (Fig. 5A). Furthermore, analysis of the change in the expression of immune scMEP markers from iDC to mDC, revealed significant downregulation of CD86 and HLA-DR in melanoma patients with a progressive decrease in worse outcome group DC (Supplementary Fig. 4C).

Differential expression analysis for scMEP markers between HD vs bad (top) and good (bottom) outcome groups in Fig. 5B revealed significant upregulation of metabolic TCA/ETC regulators (CytC, ATP5A) along with glutaminolysis enzyme glutaminase (GLS) in HD as compared to melanoma patient mDC. This was consistent with the overall observed increase in mitochondrial and glutaminolysis metabolic functions as measured by Seahorse and SCENITH assays. PPARγ co-activator-1α (PGC1α) was used in the scMEP panel to monitor overall mitochondrial biogenesis and dynamics[48,49]. Consistent with increased proton leak across the membrane as measured by Seahorse assay PGC1α was downregulated in melanoma DC suggesting their impaired biogenesis may contribute to the observed metabolic dysfunction as compared to HD (Supplementary Fig. 5A). Glutathione synthase (GSS) is involved in ROS detoxification[50] and its expression is significantly lower in worst outcome mDC compared to HD. Because deficiencies in mitochondrial glutathione have been implicated in increased ROS production[51], decreased levels of GSS in melanoma mDC may contribute to the increased protein leak observed by Seahorse measurement (Fig. 2D). PD-L1 was significantly downregulated in melanoma patient mDC (Fig. 5B). While the differential expression of multiple scMEP factors overlapped between good and bad outcome groups, mDC from bad outcome groups exhibited more pronounced and significant changes in scMEP marker differences from HD (Fig. 5B). Additional comparisons reveal that numerous TCA/ETC scMEP molecules showed reduced expression in melanoma patient mDC (Fig. 5C, Supplementary Fig. 5A). The lactate transporter MCT1, which was the most robust marker correlating with glycolytic metabolism in monocyte-derived mDC in our recent study[17] exhibited an increased expression trend in melanoma mDC (Fig. 5C). Consistent with reduced FAO capacity, β-oxidation pathway enzyme HADHA exhibited a decreased expression trend in melanoma mDC (Fig. 5C).

To determine whether immune scMEP marker expression correlated with ex vivo melanoma antigen-specific T-cell responses, we observed that CD11c and PD-L1 were significantly elevated in mDCs

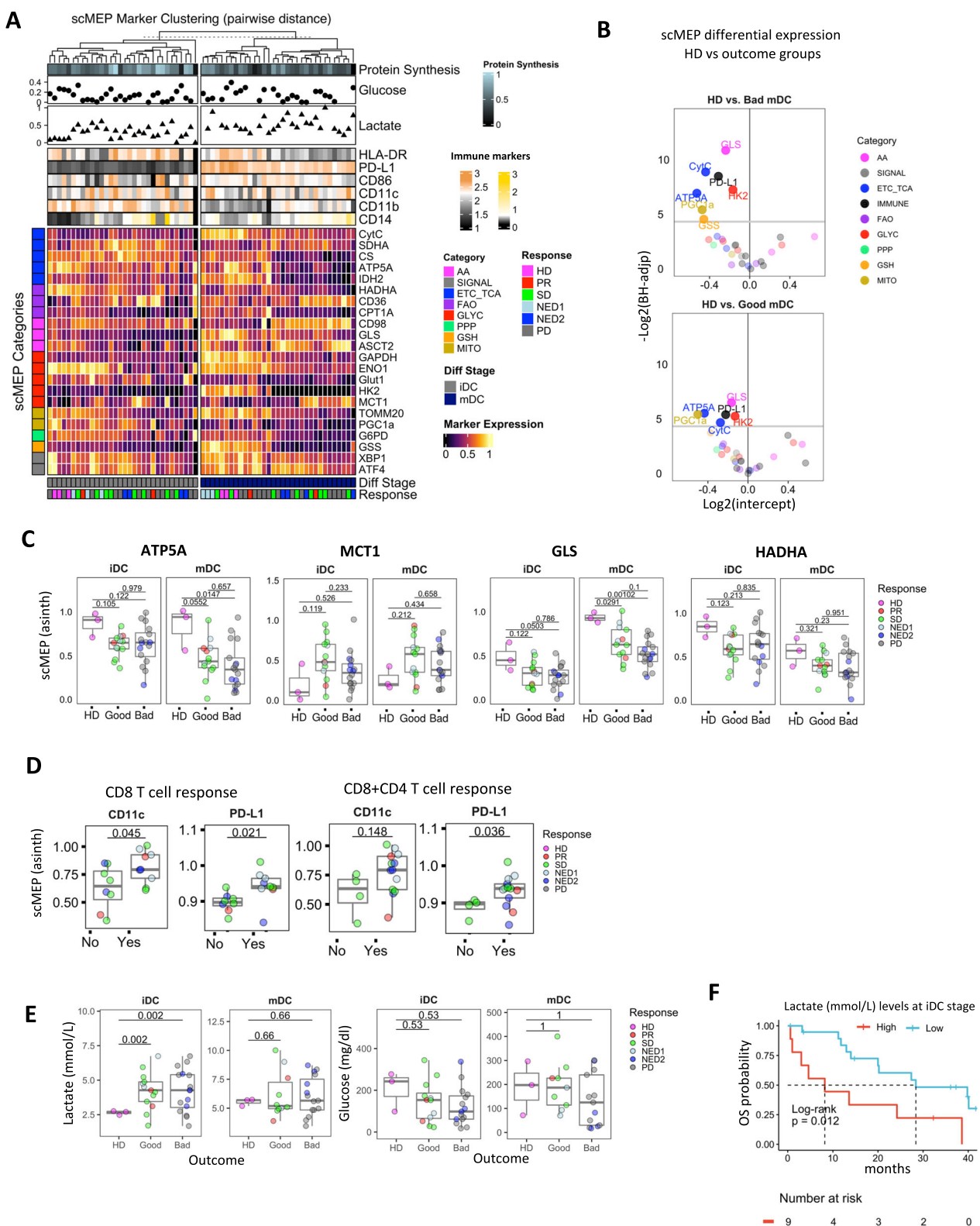

from patients with positive CD8 and combined CD8 + CD4 T cells responses (Fig. 5D).

## Increased lactate secretion inversely correlates with OS in melanoma patients

The functional implication of increased MCT1 expression and glycolytic metabolism was further demonstrated by measuring the byproduct of glycolytic pathway activity, lactate, as well as glucose in DC supernatants.

Lactate levels correlated with MCT1 scMEP expression and were significantly increased in culture media from melanoma patient-derived cells particularly at the iDC differentiation stage (Supplementary Fig. 5B, Fig. 5E). Lactate levels inversely correlated with the glucose concentration in media, indicating increased glucose

**Fig. 5 | Metabolic regulome profiling by scMEP with glucose and lactate measurements. A** Heatmap of heathy donor (HD, $n = 3$), good (PR/SD/NED1, $n = 13$) and bad (PD/NED2, $n = 17$) and melanoma iDC and mDC based on median arcsinh transformed expression values for metabolic scMEP markers. Bottom heatmap annotations include DC stages and clinical groups. Quantitation of protein synthesis levels, point annotations representing lactate and glucose supernatant measurements and expression values for DC immune signatures are displayed in the top annotations. Row annotations represent classes of scMEP markers within respective metabolic pathways. **B** Volcano plots show linear regression intercepts for differential median scMEP marker expression by fold change (logFC) and BH FDR adjusted $p$-value ($-\log2(BH\text{-adj.p})$) comparing HD vs Good (PR/SD/NED1, $n = 13$) and Bad (PD/NED2, $n = 17$) outcome groups respectively. Horizontal solid lines indicate significance threshold (BH-adj.p = 0.05) with vertical dotted lines marking fold change ≤1.5 with colors representing scMEP metabolic pathway. **C** Box plots represent differences in expression of median scMEP expression profiles for metabolic markers in mDC between heathy donor (HD, $n = 3$), good (PR/SD/NED1, n = 13) and bad (PD/NED2, $n = 17$) clinical groups. **D** Median scMEP marker

expression stratified by absence (No) or presence (Yes) of positive CD8 and combined CD8 + CD4 IFN-γ T-cell responses specific to melanoma antigens ($n = 17$). **E** Glucose and lactate measurements from DC culture supernatants in heathy donor (HD, $n = 3$), good (PR/SD/NED1, $n = 13$) and bad (PD/NED2, $n = 17$) clinical groups. Of note glucose level measurement increase in the media between d3 and iDC stage is due to media change at day 3. Three technical replicates from 3 donors are presented with error bars indicating standard deviation. Multiple comparisons were calculated via one-way ANOVA with Tukey's *post-hoc* test. **F** Kaplan–Meier survival analysis of OS with indicated log-rank test comparing the inferior survival benefits of increased lactate in supernatants from melanoma patient-derived iDC. In (**C**, **D**–**E**) Box plots indicate 1st, 2nd and 3rd quartile; whiskers indicate minimum and maximum. Multi-group comparisons in (**C**–**E**) were tested by one-way ANOVA with Tukey's *post-hoc* test. In **D**, Shapiro–Wilk test was used to assess data normality, Two-tailed Wilcoxon signed-rank test (non-normal data) and Two-tailed Student's *t*-test (normal data) was used for statistical analysis. Source data are provided as a Source Data file.

---

consumption (lowered glucose in media) by melanoma DC (Supplementary Fig. 5C). We observed a significant increase in the fraction of glucose being converted to lactate in the melanoma iDCs and no difference between outcome groups was observed (Supplementary Fig. 5D). Gene expression levels of additional MCT family transporters did not show a correlation with lactate levels. Further, MCT1 was the highest expressed transporters, supporting its importance in mDCs (Supplementary Fig. 5E, F). Lactate is a potent immunosuppressive metabolite in the context of oncogenesis and inflammation and has been considered a predictive or prognostic biomarker of clinical response[52]. Kaplan–Meier (KM) survival analysis comparing levels of lactate in iDC culture supernatant confirmed that increased lactate secretion by DC significantly correlated with inferior OS rate of patients (Fig. 5F).

### DC cytokine expression analysis

To gain further insights into the protein secretion profiles of the patient mDC, culture supernatants were tested for cytokines, chemokines and growth factors, as well as secreted or shed checkpoint and costimulatory molecules (HD ($n = 4$) vs. melanoma patient ($n = 23$)) (Fig. 6). A heatmap showing the cumulative data clustered by clinical outcomes and indicating CD4+ and CD8+ T-cell response results is in Fig. 6A. Patients with PD show the least secretion of any of the proteins measured. The statistical significance of these results with clinical outcome indicates that DC secreting higher levels of many of the analytes associates with positive outcome (Fig. 6B). While it is surprising that the T and NK cell growth and survival factor IL-15 was associated with poor outcome, this may be due to the very low levels of this protein measured overall, and particularly high expression in a single PD patient culture.

In many cases, melanoma patient DC secreted much lower levels of analytes than HD, regardless of clinical outcome (HGF, IL-12p70, TNFA). While IL-12p70 has been a major focus for DC due to its promotion of Th1/Tc1 immunity, in this and other studies[40], the amount of IL-12p70 secreted by DC did not correlate with T-cell response or clinical outcome. Other analytes showed a trend of being highest in HD, then good outcome and lowest in bad outcome patient DC (CXCL13, eotaxin, IL-23, IL-31, IL-5, MCP-1, MIG, sCD40L, TIM3, TRAIL). These proteins are associated with multiple response profiles, including Th1, Th2 and myeloid cell trafficking. There is also a subset of analytes which are strong in both HD and good outcome patient cells, but reduced in bad outcome patients (IFNα, IL-18, IL-1α, IL-21) all of which have type 1 skewing and antitumor immunity activity. Next, we assessed DC metabolism and protein secretion profiles and show that SCENITH-derived glucose dependence of mDC exhibited significant inverse correlations with secretion of IDO, BTLA, and GITR. Additionally, LAG3, APRIL and TNFß exhibited inverse correlations with

glycolytic capacity and maximal oxygen consumption of mDCs respectively (Fig. 6C), suggesting that metabolic state impacts the immune-related protein secretion profile of DC.

### Immune phenotype alteration of circulating monocyte and DC subsets differentiate cancer patients from HD and elevated ILT3 and PD-L1 expression associate with worse prognosis

Given the impact of the metabolic state of DC vaccines on immune phenotype and clinical outcome of vaccinated patients, it was critical to determine whether the vaccine progenitor cells, the circulating monocytes, were already impacted at baseline in melanoma patients. To characterize metabolic states of circulating monocytes as well as DC subsets in melanoma patient blood, we combined SCENITH with the high-dimensional immune-phenotyping panel based on the most recent classifications of monocytes, plasmacytoid and conventional DC subpopulations[53,54] (Supplementary Fig. 5A, Supplementary Table 3).

We observed a significant increase in plasmacytoid CD123 + DC (pDC) and decrease in conventional CD5 + cDC2 and CD14-DC3 frequencies in melanoma patients (Supplementary Fig. 6A). There were no significant changes in cDC1 frequency while classical (cMo) and intermediate (iMo) showed a trend of a decrease and increase in frequency, respectively (Supplementary Fig. 6A). Clustering analysis of the combination of inflammatory, classical and more recently defined lineage markers revealed specific patterns of key molecule expression, and delineated circulating monocyte and human DC subsets (Fig. 7A). PCA analysis using these 21 immune markers revealed cluster separation based on monocyte, plasmacytoid and conventional clusters (Fig. 7B). An overlay of the clinical responses showed separation of HD from melanoma groups, particularly in the monocyte and plasmacytoid clusters. These data suggest that the circulating myeloid compartment of melanoma patients may be significantly different from that of healthy donors. Further evaluation of immune marker expression revealed that PD-L1 (on the majority of populations) and CD36 on ncMo and pre-DC were significantly increased on heathy donors (Fig. 7C).

Additional comparisons showed significant upregulation ILT3 in iMo and cDC1s with downregulation of PD-L1 and CD206 in iMo in non-responders (Fig. 7D). Cox's regression analysis further demonstrated that increased expression of ILT3 on selected monocyte and conventional DC subtypes was significantly associated with decreased OS in melanoma patients. In contrast, CD11c expression on monocyte lineages is a protective predictor, as well as PD-L1 expression on conventional subsets, which is associated with improved progression free survival in melanoma patients (Fig. 7E). For these circulating cells, PD-L1, which is upregulated by DC maturation, may indicate a positive activation, while ILT3 is a negative functional checkpoint in these cellular subsets.

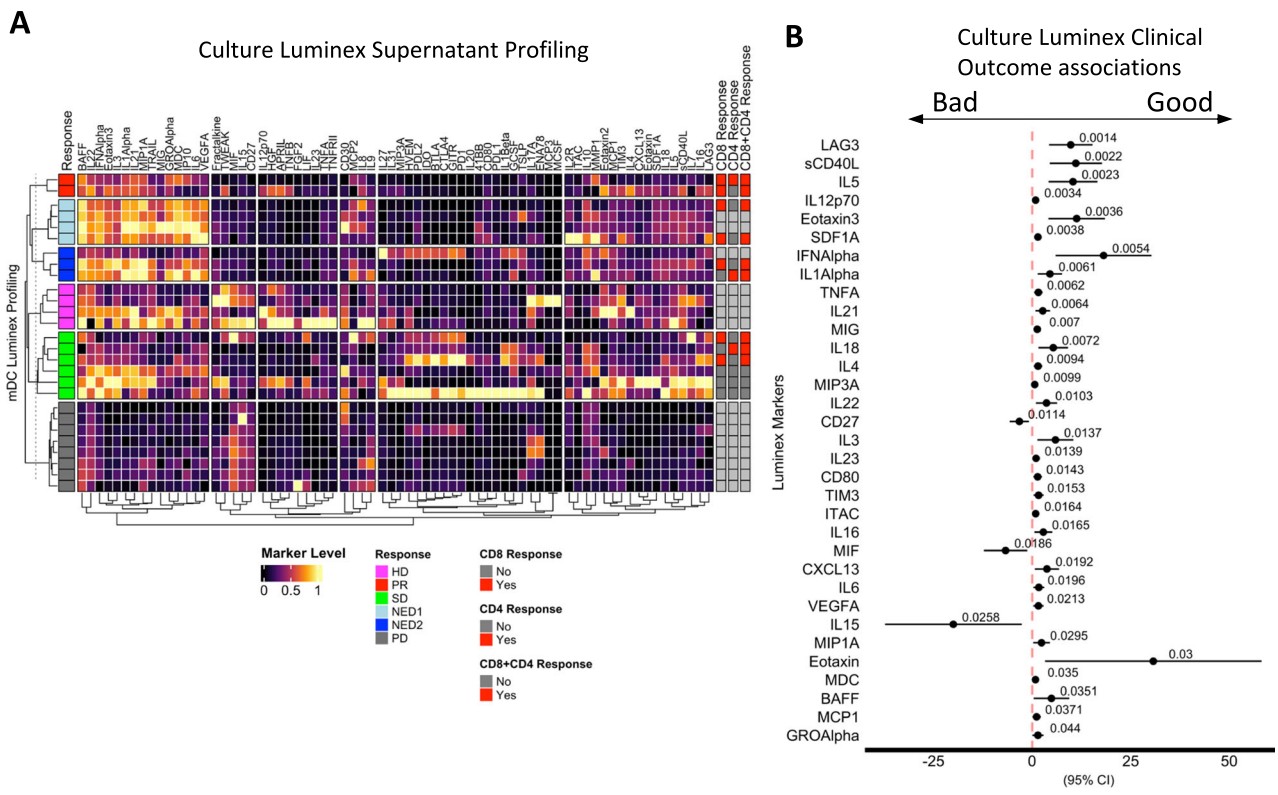

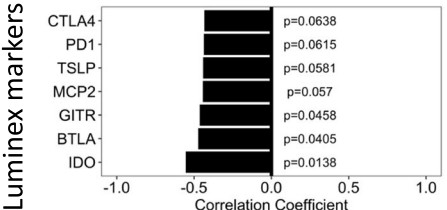

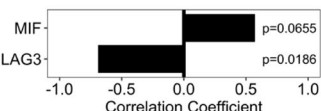

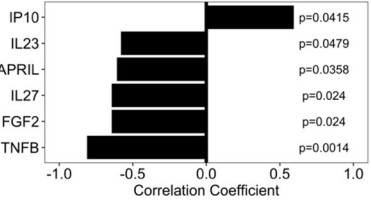

**Fig. 6 | Luminex mDC culture supernatant profiling. A** The Human Checkpoint 14-plex and immune profiling 65-plex assay kit (Thermo-Fisher ProcartaPlex) were used to measure immune-modulatory molecules in mDC culture supernatants from 4 healthy donors and 27 melanoma patients. Row labels include HD and patient response indications and absence (No) or presence (Yes) of patient-derived (MA)-specific CD8, CD4, combined CD8 + CD4 IFN-γ T-cell responses. **B** Forest plot summarizing linear regression analysis for immune-modulatory chemokine/cytokine levels with clinical outcome associations. Intercept points, *P*-values and 95% confidence intervals are indicated (HD, *n* = 4, Melanoma, *n* = 27). **C** Bar plots represent Spearman correlation coefficient (R) and *p*-values of mDC culture supernatants from 27 melanoma patients correlating with SCENITH metabolic parameters Glucose Dependence, and Seahorse assay glycolytic capacity and maximal oxygen consumption rate. Source data are provided as a Source Data file.

## Monocyte/myeloid circulating populations exhibit metabolic changes and immune differences between HD and melanoma patients

Finally, we performed single cell analysis of concurrently tested oligomycin-treated SCENITH samples. Dimensionality reduction based on 21 immune markers using tSNE maps showed visual separation of monocyte and DC populations (Fig. 8A). Furthermore, overlay of puromycin expression quantiles from these oligomycin-treated samples enabled us to visualize the single cell glycolytic and mitochondrial states of these distinct populations (Fig. 8A top).

Analysis of cell proportions within each metabolic quantile demonstrated that circulating classical to non-classical monocyte populations exhibit shift from glycolytic to mitochondrial metabolism. Within the conventional DC, cDC1s have the highest proportion of cells with mitochondrial respiration quantile (Fig. 8A bottom). The majority of cells in the pre-DC, CD5 + cDC2 and DC3s populations utilize glycolytic metabolism. We compared cell subtype metabolic profiles between HD and melanoma patients stratified by clinical outcome. cMo and ncMo exhibited significant decrease in mitochondrial dependence with decreased trends in FAO and glutaminolysis (Fig. 8B). iMo

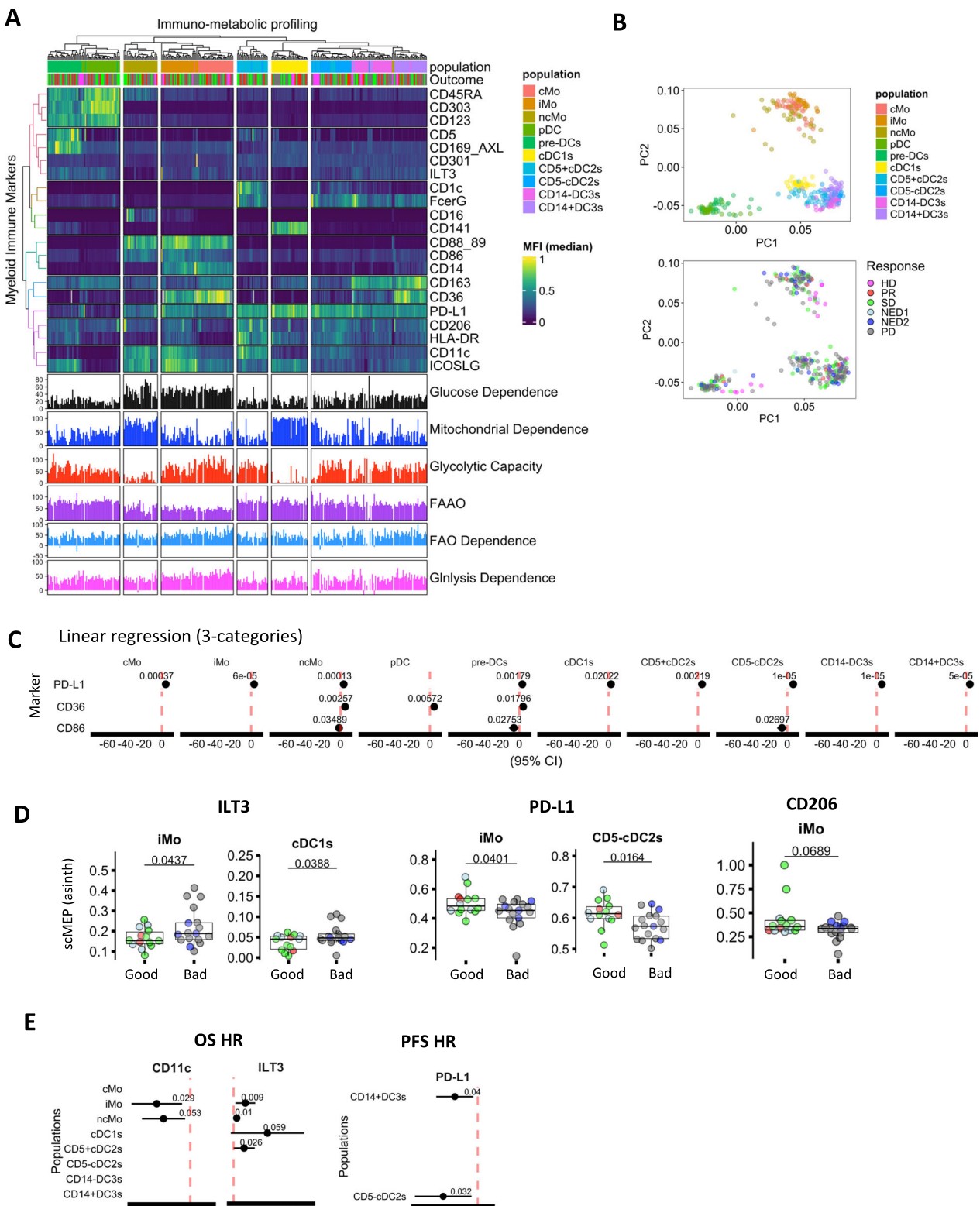

**A** Immuno-metabolic profiling

**B**

**C** Linear regression (3-categories)

**D**

**E** OS HR    PFS HR

did not reveal significant metabolic changes while pDC and pre-DC showed a trend towards progressive decrease in glutaminolysis dependence in non-responders (Fig. 8B, Supplementary Fig. 6B). Glucose dependence was significantly reduced in conventional cDC1s and CD14 + DC3s, while both cDC2 subtypes exhibit decreased mitochondrial dependence.

Based on the overall assessment of metabolic profiles, amino acid metabolism was more severely affected in melanoma patient circulating myeloid subsets compared to FAO. We further demonstrate that the effects of underlying metabolism on immune phenotypes is also reflected at the baseline circulating myeloid cell level. Intermediate monocyte populations (iMo) were the closest predictors of glycolytic capacity and mitochondrial dependence in cultured mDC (Supplementary Fig. 7A). In contrast to ncMo, mitochondrial dependence and glycolytic capacity in cMo (the majority of the monocytes) and iMo were positively corelated, while FAO and glutaminolysis

**Fig. 7 | Clinical correlations for immune and metabolic phenotypes of circulating monocyte/myeloid and DC populations from melanoma patients.**
**A** Integrated clustering heatmap of median MFI expression profiles for circulating myeloid/DC subtype populations (HD, $n = 3$, melanoma, $n = 30$) (gating strategy is shown in Supplementary Fig. 5A) profiled by SCENITH (marker/antibody information is available in Supplementary Table 3). Percentual metabolic parameters are shown underneath, with response groups and population labels presented on the top of the heatmap. **B** Principal component analysis (PCA) for HD ($n = 3$), and melanoma ($n = 30$) circulating myeloid/DC populations based on lineage and inflammatory marker expression. Clinical response indications are overlaid in the bottom PCA graph. **C** Forest plots indicate linear regression results for circulating myeloid/DC populations marker expression associations with (HD, $n = 3$), good

(PR/SD/NED1, $n = 13$) and bad (PD/NED2, $n = 17$) outcome groups. Intercept points, P-values and lines denoting 95% confidence intervals are indicated. **D** Box plots represent differences in expression of median scMEP expression profiles for metabolic markers in myeloid/DC populations between good (PR/SD/NED1, $n = 13$) and bad (PD/NED2, $n = 17$) clinical groups. Box plots indicate 1st, 2nd and 3rd quartile; whiskers indicate minimum and maximum. Statistical significance between good (PR/SD/NED1, $n = 13$) and bad (PD/NED2, $n = 17$) outcome groups was determined using Two-tailed Student's t-tests. **E** Univariate Cox regression analyses for marker expression levels and overall and progression free survival ($n = 30$) with P-values and 95% confidence intervals indicated Source data are provided as a Source Data file.

dependence were inversely correlated with mDC metabolic profile (Supplementary Fig. 7B).

However, it is important to consider that there is variable expression of the immune markers in different circulating myeloid subtypes (Fig. 8C). HLA-DR expression particularly in iMo, pre-DCs, cDC1s, cDC2s does not seem to be impacted by the underlying metabolic state. In contrast, ILT3 and PD-L1 levels on most circulating myeloid monocytes and DC subtypes is differentially expressed on glycolytic and mitochondrial populations respectively (Fig. 8C).

## Discussion

Overall, this study analyzed the transcriptomic, phenotypic and metabolic profiles of mDC from 35 subjects enrolled in a Phase I study of autologous DC vaccines in late-stage melanoma[40]. Multiple platforms were utilized to identify correlations and aspects of DC function which were important for overall survival. Microarrays revealed differences in immune gene signatures between HD and patient DC including increased MHC class I presentation, antigen processing and CXCR chemokine pathway in HD donor mDC. In addition, TGFβ, NLRP3 inflammasome, Oncostatin M and VEGF/VEGFR signaling pathway was enriched in melanoma DC, which has been shown to be inhibitory to DC maturation and function[55]. Relevance of metabolic alterations in melanoma mDC was seen in gene signatures involved in the TCA cycle and electron transport chain/OXPHOS in HD and FA/phospholipid metabolism and PPAR pathways.

Seahorse metabolic flux functional testing identified increased glycolytic capacity and basal glycolysis as important negative functional skewing in poor outcome patients. Along with increased glycolysis, we observed reduced maximal exogenous FAO along with increased proton leak and reduced ATP-linked respiration in melanoma DC. Increased proton leak and increased reactive oxygen species production has been previously associated with age-related mitochondrial dysfunction, with inhibitory effects to phagocytosis and T-cell MHC cross-presenting activity of aging DC[56].

Given the heterogeneity of the patient DC, the population-based comparisons in molecular pathway identification as well as changes in mitochondrial OXPHOS pathways between outcome groups were more challenging to dissect. The overall decrease in gene enrichment profiles relating to mitochondrial TCA/ETC signatures in HD as compared to melanoma patient mDC did not correlate with functional Seahorse metabolic assays. This suggest that transcriptomic profiling of metabolism may not reflect functional metabolic states due to a lack of consistent direct correlation between gene expression and protein level modifications as well as the heterogeneity of the cell populations.

Therefore, to validate the population-based seahorse measurements and better capture immune phenotypes in conjunction with metabolic states associated with the heterogeneous nature of in vitro patient-derived mDC cultures, we employed SCENITH and scMEP cytometry-based approaches. We previously employed these methods to demonstrate the importance of mitochondrial dependence in monocyte-derived DC differentiation and that elevated glycolytic metabolism along with increased mTOR:AMPK phosphorylation ratio

reflects metabolic hyperactivation of tolerogenic DC with less-well matured tolerogenic DC phenotype[17]. While glycolytic metabolism is a hallmark of mBMDC activation, this phenomenon does not directly translate to human DC[14–17]. While often associated with metabolism favoring long-lasting or quiescent immune cells, TCA/OXPHOS play more important role in inflammatory activation of human DC than was previously appreciated[47,57]. Here we further show that elevated glycolysis with reduced mitochondrial dependence, glutaminolysis and FAO is a hallmark of melanoma mDC compared HD. Decrease in mitochondrial dependence in melanoma DC was closely associated with significant reduction in scMEP OXPHOS markers CytC, ATP5A and IDH2. Based on these results, we suggest that mitochondrial dependence is an important parameter of DC maturation status and may also be a valuable biomarker of clinical response as higher mitochondrial dependence in patient mDC was significantly associated with longer OS and PFS rate.

An important role of amino acid metabolism was implicated in recent studies, in which inhibition of glutaminolysis was linked to suppression of Tfh13 polarization by DCs in allergic asthma[58] and expression of amino acid transporters was required for mTORC1 activation and effector function of pDC[59]. Patente et al. also demonstrated that glutamine fuels a TLR-stimulation dependent increase in OXPHOS metabolism and mitochondrial content in pDC[60]. Here we observed an importance for amino acid metabolism, specifically showing that both GLS expression and functional glutaminolysis dependence were significantly reduced in melanoma patients, with a progressive decrease in worse outcome group DC. The goal of DC vaccination is to induce or expand functional and long-lived tumor-specific immunity[61] and we previously showed that CD8+ T cells were critical to clinical outcome (PFS and OS) as well as vaccine-encoded antigen-specific T-cell responses[40]. Surprisingly, metabolic parameters in melanoma DC were less predictive of antigen-specific T-cell responses, although a trend towards higher mitochondrial dependence associating with positive T-cell responses was observed. T-cell activation and response is a highly dynamic and context-depended process and evidence suggests that DC utilize different metabolic states to drive polarization of different Th cell subsets[3]. Therefore, the steady state metabolic profiling performed in our study may not be suitable as an accurate predictor of the T-cell priming responses. We also hypothesize that metabolism may affect other aspects of DC biology including survival and migratory capacity to lymph nodes that also plays a large role on their efficacy to mount successful immune response in patients.

To support the importance of mitochondrial metabolism in maturing DC we blocked p-AMPK (a positive regulator of mitochondrial metabolism) and show that reduced pAMPK resulted in the reduced expression of several immune makers including HLA-DR, CD86, PD-L1 and CD206 in HD mDC. Because we also observed reduced glucose uptake along with lower extracellular lactate levels we speculate that blockade of p-AMPK has broad impact on mDC metabolism. Additional studies are needed to precisely link and separate the effects of p-AMPK blockade on mitochondrial respiration and glycolysis in mDC.

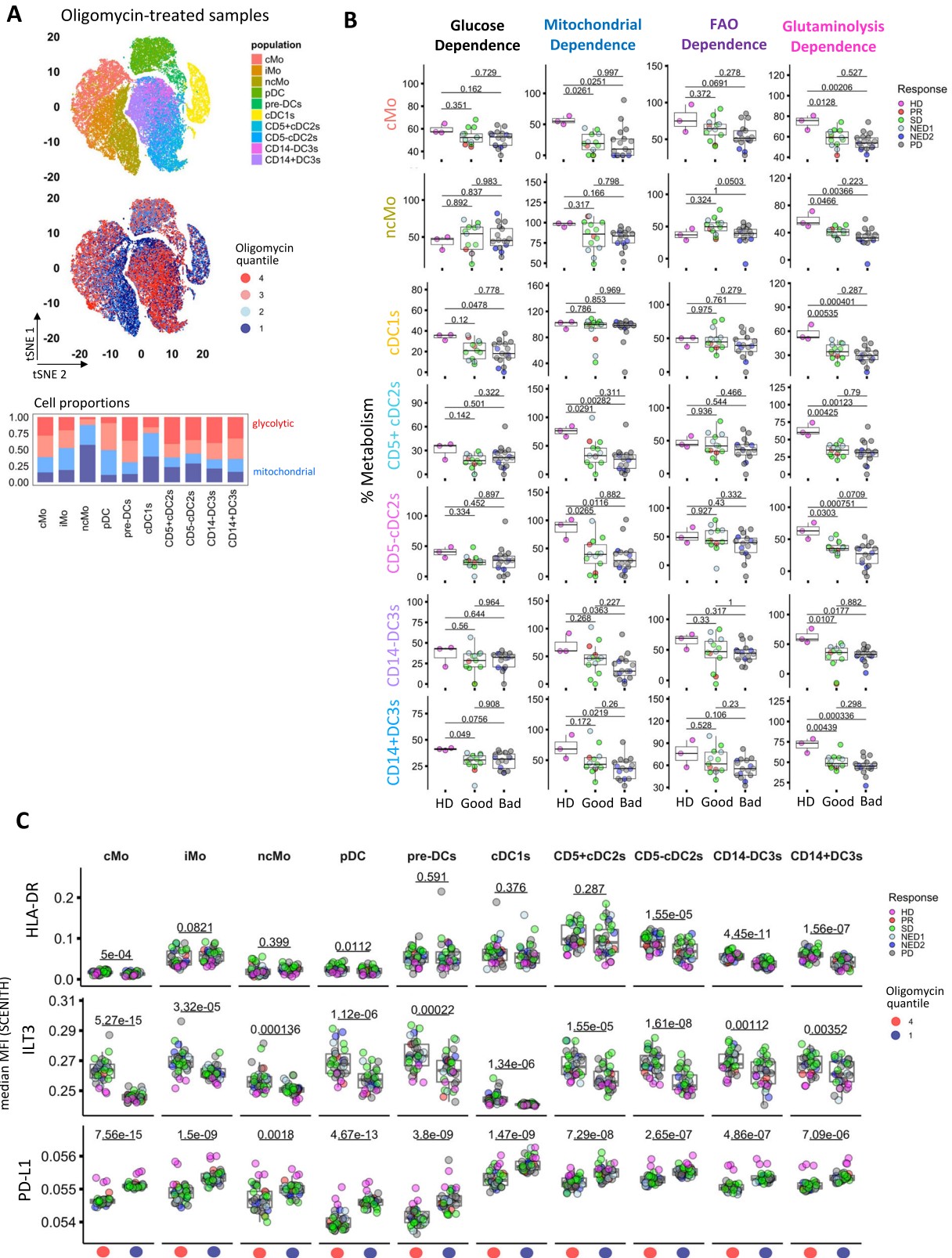

Using more in depth analysis of oligomycin-treated single-cell experiments, we demonstrated that proportions of mDC with elevated of glycolytic dependence increases in bad outcome groups. We also observed distinct patterns of DC immunophenotypic marker expression between glycolytic and mitochondrial-dependent mDC populations. While the glycolytic populations with increased pmTOR:AMPK ratio exhibited overall increases in surface expression of ILT3, HLA-DR,

CD86, PD-L1, CD206 and CD40, these patient mDC exhibited more heterogeneity and were more distant from HD cells in clustering analysis based on immune parameters.

In contrast, mDCs with highest mitochondrial dependance exhibited closer clustering with HD which suggested that these are more uniform and immunologically similar populations. We speculate that the aberrant increase in glycolytic metabolism may reflect

**Fig. 8 | Profiling the effects of metabolic states on immune phenotypes of circulating monocyte/myeloid and DC populations in HD and melanoma patients.** A T-distributed stochastic neighbor embedding (tSNE) based on lineage/inflammatory markers (indicated in Supplementary Table 3) was used to visualize multidimensional separation of Oligomycin-treated circulating myeloid/DC subtype populations from HD and melanoma patients. **B** Box plots represent differences in SCENITH metabolic parameters in circulating myeloid/DC subtypes between heathy donor (HD, $n = 3$) vs. good (PR/NED1/SD, $n = 13$) and bad (PD/NED2, $n = 17$) response groups. Pairwise comparisons against a HD reference group in were calculated using Two-tailed Student's t-test with Holm-Bonferroni correction. **C** Box plots represent comparisons of median MFI expression profiles for circulating myeloid/DC subtype populations between glycolytic (red) and mitochondrial-dependent (blue) oligomycin quantiles ($n = 30$). In **B**, **C** Box plots indicate 1st, 2nd and 3rd quartile; whiskers indicate minimum and maximum. Statistical significance between outcome groups was determined using Two-tailed Student's t-tests. Source data are provided as a Source Data file.

a transitional and/or pathological state similar to recently proposed pathologically active glycolytic state in monocytes from tuberculosis patients, which limited generation and migratory capacities of monocyte-derived DCs[62]. This chronic state with aberrant immune maturation does not resemble the highly glycolytic phenotype of maturation-deficient tol-DC or p-AMPK inhibitor treated mDC and future studies should be conducted to elucidate the functional implications of this metabolic state in generation of melanoma DC vaccines[16,17,23,24]. Collectively, these data further support that the underlying metabolic states can influence immune phenotypes of maturing DCs. Distinct DC surface markers can be prone to altered expression based on the glycolytic or mitochondrial polarization of mDC.

scMEP quantification of metabolic enzymes, transporters and signaling nodes we show that changes in the metabolic regulome and coordinate activation of multiple metabolic pathways in mDC differentiation and maturation are important correlates of clinical outcome.

The most correlated scMEP marker for increased glycolytic metabolism in melanoma DC is the MCT1 lactate transporter. As a member of the monocarboxylate transporters (MCT) 1–4 family of receptors[63], MCT1 facilitates both import and export of lactate depending on the pH gradient[64]. Lactate, a byproduct of cellular glycolysis, functions as immuno-inhibitory molecule in the context of inflammation, T cells activation suppression[23] and tumor microenvironment[65].

MCT1 lactate transporter and lactate in cultures were increased in poorest outcome patient cells, supporting elevated glycolysis in melanoma DC. Furthermore, elevated lactate levels in supernatants was inversely associated with OS in our melanoma cohort. This relates to our recent analysis of the impact of the immune-suppressive tumor-derived alpha fetoprotein (AFP) on DC function[66]. In this study, we identified the mechanism by which tAFP reduces the immune stimulatory activity of DC, and also show increased glycolysis, decreased mitochondrial dependence and increased lactate secretion in poorly stimulatory DC impacted by cancer cell-derived factors (AFP). Given our other recent study in tolerogenic HD DC, these three studies support a reduction in broad metabolic capability to use multiple fuels for cellular activity towards a skewed use of glucose as fuel and lactate secretion as a common mechanism of reduced DC stimulatory activity in cancer. Our protein secretion analysis showed several correlations with DC metabolic parameters. LAG3 is an immunosuppressive checkpoint molecule known to block T-cell activation, however as a soluble ligand, LAG3 was shown to bind to MHC class II, and promote maturation and CD8 T-cell antigen presentation by DC[67,68]. Consistent with these reports, we hypothesize that soluble LAG3 may be a cell surface indicating optimal maturation of patient-derived mDC, as its level was inversely correlated with glycolytic mDC which exhibited impaired differentiation. A group of immune suppressive molecules CTLA4, PD-1, IDO, BTLA implicated in inhibition of CD4 T-cell proliferation by DCs[69] were decreased in mDC cultures that exhibited higher glucose dependence, implying that lower glucose dependence is an immunosuppressive profile of DC vaccines. The functional role of glucose as well as other energy sources including fats and amino acids in immune phenotype and protein secretion requires further dissection. IP-10 has been implicated in regulatory DC-mediated recruitment and inhibition of Th1 cell proliferation[70] and we observed its positive correlation with maximal oxygen consumption rate along with inverse correlation with TNFSF receptor family members APRIL and TNFB as well as IL-12 family members IL-23 and IL-27, known to influence B cell activation and Th1/2 cell responses[71,72]. These observations from DC cultures are hypothesis generating, and the functional link between DC metabolic state and protein secretion profiles requires further experimental testing.

Precise understanding of metabolic requirements of DC subsets in the blood has been limited largely due their low frequencies and technical limitations, and we herein now describe metabolic differences in distinct circulating DC subtypes in heathy and melanoma disease settings. CD141+ cDC1s characterized are particularly efficient at MHC-I restricted exogenous antigen cross-presentation to cytotoxic CD8+ T cells, and based on oligomycin quantile analysis, cDC1s have the highest proportion of cells with mitochondrial respiration quantile. While effective at phagocytosis mediated MHC-II antigen presentation, cDC2 subtypes are involved in regulating mucosal $T_H17$ immunity[73], antitumor T-cell responses and cytokines and chemokines production directing inflammatory CD4 T-cell polarization[74]. These conventional cDC2 subsets exhibited a largely glycolytic profile. DC3s are a unique DC lineage sharing characteristics of both cDC2s and monocytes and are distinguished using CD14, CD5 and hemoglobin scavenger receptor CD163 markers[67] and were recently shown to be less susceptible to apoptosis compared cDC2, with elevated frequencies during anti-viral inflammatory response in SARS-CoV-2 infection[68]. Overall metabolism of DC3s was also primarily glycolytic. In comparative analysis between HD and clinical response groups, we show that circulating myeloid cells in melanoma patients are also skewed in metabolic function.

Across cell types, a decrease in glutaminolysis dependence was a hallmark feature of multiple circulating DC subsets in melanoma patients with progressive decrease in worst prognosis patients. While FAO dependence was largely unaffected, cMo, ncMo and both cDC2 subtypes exhibited decreased mitochondrial dependence with a corresponding increase in glycolytic capacity. Based on our results, glycolytic capacity and mitochondrial dependence in iMo was most predictive of these metabolic parameters in cultured mDC, and the majority population of cMo also showed a trend towards being a predictive biomarker of cultured DC metabolism. Many of the cell surface examined were increased or decreased relative to HD, however lactate secretion should be tested as a biomarker. We also aimed to determine whether any of the metabolic parameters of the monocyte precursors in periphery are correlative and indicative of their metabolic state in cultured mDC. Based our results, metabolic profiles of iMo showed the closest correlation trends of the metabolic parameters in cultured mDC. FAO dependence was the only parameter showing significant correlation, but the implication of using metabolic profiles of peripheral monocyte precursors in predicting metabolism of mDC will require further validation. Many of the cell surface examined were increased or decreased relative to HD, however lactate secretion of should be tested as a potency biomarker.

We did not detect correlations with CD8+ and CD4+ T-cell responses with circulating DC metabolism, however we observed an increased expression of the checkpoint receptor ILT3 on selected monocyte and conventional DC subtypes in melanoma patients, which significantly associated with decreased OS. Because ILT3 is an inhibitory receptor[69], negatively regulating activation of APCs, we suggest

that its elevated expression may contribute to reduced T-cell stimulatory potential of melanoma patient-derived mDC. Collectively we show that circulating myeloid cells in melanoma patients are also skewed in metabolic function.

Here, we show that the metabolism of the DC is significantly associated with OS, and also weakly correlated to PFS. It may be that ex vivo DC metabolism is a reflection of the circulating myeloid cellular compartment metabolic functionality and overall immune health of the patient, and the vaccine T-cell response is less directly tied to DC metabolism, but is a separately significant mechanism of successful antitumor immunity development. Future studies focusing on functional consequences of the immune-metabolic deficiencies in the distinct DC blood subtypes may be conducted to better understand their involvement in the cancer state.

The therapeutic efficacy of ex vivo cultured DC vaccines derived from such metabolically skewed monocytic cells may require metabolically defined culture conditions to create more effective vaccines capable of metabolic flexibility, and subsequently of inducing more effective antitumor T-cell responses and positive clinical outcomes. The metabolic defects in circulating monocytes and DC subsets we observed here could reduce the efficacy of vaccines primarily delivering antigens for presentation by endogenous APC. Mechanisms by which cancer and cancer therapy may induce these changes are under investigation.

## Methods

Specimens were obtained from a Phase I, single site study (NCT01622933) designed to evaluate the toxicity and immunologic and clinical responses from autologous DC transduced with the tyrosinase, MART-1 and MAGE-A6 genes in 35 subjects with recurrent, unresectable stage III or IV melanoma (M1a, b, or c), or resected stage IIIB-C or IV melanoma. The endpoints were local and systemic toxicity, generation of immunity to immunizing antigens and epitope spreading, and clinical response. Enrollment occurred from 9/2012–11/2015, after institutional scientific and IRB approvals (UPCI #09-021) and with informed consent. We confirm that our research complies with all relevant ethical regulations; the clinical trial was conducted with full IRB approval at the University of Pittsburgh. We also confirm that the study design and conduct complied with all relevant regulations regarding the use of human study participants and was conducted in accordance with the criteria set by the Declaration of Helsinki.

### In vitro monocyte-derived DC generation

Leukapheresis cells were elutriated into myeloid and lymphoid fractions. The myeloid cells were cultured for 5 days to generate immature Dendritic Cells (iDC) from cryopreserved elutriated healthy donor and patient monocytes using 1000 U/mL GM-CSF (Genzyme and Sanofi) and IL-4 (Cell Genix).in DC medium (Cell Genix). DC were matured using rhIFNγ (1000 U/mL) (Actimmune and R&D Systems) and LPS (250 ng) (Sigma Aldrich) in DC medium for 24 h. Immature and matured Dendritic Cells were harvested. Viability was analyzed using a Trypan Blue viability dye. Dorsomorphin (Selleckchem,7306) was added at iDC stage together with IFN-γ/LPS for 24h.

### Microarray and Gene Expression Analysis (GSE157738) and (GSE111581)

Total RNA from 5×10e6 iDC, mDC and vaccine DC was isolated using RNAlater (Qiagen). HUGENE 2.0 ST arrays (Affymetrix) was used for gene expression analyses.

Differential gene expression was analyzed using limma (Version 3.38.3) with weights generated by the voom function[70,71]. A log2 fold change of 2 and FDR-adj.p-value threshold of 0.05 was used to determine statistical significance. Web-based tool gProfiler[72] was used for pathway analysis of significantly up and downregulated gene sets. Gene set enrichment analysis (GSEA) was conducted using gene sets

from the Molecular Signature Database (MSigDB, Version 6.2) in the C2 curated gene category (2005, PNAS 102, 15545–15550). Plots were generated using the R package ggplot2 (Version 3.1.1) and the javaGSEA application (version 3.0). Molecular interaction networks were determined and visualized using the Cytoscape (version 3.7.0)[75]. Relative enrichment of gene sets across samples was performed using the GSVA R package (version 1.48.2).

### Metabolic assays

Metabolic assays were performed as described in Santos et. al.[76]. Day5 immature and Day6 matured were plated at 100,000 cells/well on Seahorse culture plates. DMEM media was used, supplemented with 1% BSA, 25 mM glucose, 1 mM pyruvate, and 2 mM glutamine. The cells were analyzed using the Seahorse XFe96 (Agilent). Basal oxygen consumption and extracellular acidification rates were collected every 30 min. The cells were stimulated with oligomycin (2 μM), FCCP (0.5 μM), 2-deoxyglucose (10 mM) and rotenone/antimycin A (0.5 μM) to obtain maximal respiratory and control values. Fatty Acid Beta Oxidation was measured using the XF Palmitate Oxidation Stress Test Kit (Aligent). To measure oxidation levels, palmitate-BSA or BSA control (30 uls) was added to the wells immediately prior to running the assay. Cells were stimulated with oligomycin (2 μM), FCCP (0.5 μM), 2-deoxyglucose (10 mM) and rotenone/antimycin A (0.5 μM) to obtain maximal respiratory and control values. For both metabolic assays, the measurements were performed in triplicates. The OXPHOS and glycolytic indices were calculated as follows

$$Basal\ respiration = OCR_{pre-Oligo} - OCR_{post-RA}$$

$$Maximal\ oxygen\ consumption = OCR_{post-FCCP} - OCR_{post-RA}$$

$$Spare\ respiratory\ capacity = OCR_{post-FCCP} - OCR_{pre-Oligo}$$

$$Proton\ leak = OCR_{post-Oligo} - OCR_{post-RA}$$

$$Maximal\ respiration\ of\ exogenous\ FA = \ = (CTRL)OCR_{post-FCCP} \\ - (Palm/ETO)OCR_{post-FCCP}$$

$$Basal\ glycolysis = ECAR_{pre-Oligo} - ECAR_{post-RA}$$

$$Glycolytic\ capacity = ECAR_{post-Oligo} - ECAR_{post-RA}$$

### SCENITH™ staining and data acquisition

SCENITH™ was performed as described in refs. 17,28. SCENITH™ reagents kit (inhibitors, puromycin and anti-Puromycin antibody clone R4743L-E8) were obtained from www.scenith.com/try-it and used according to the provided protocol for in-vitro derived myeloid cells.

Briefly, $2 \times 10^6$ melanoma patient elutriated fraction 5 cells, $2 \times 10^6$ HD PBMC, or monocytic mDC cultures ($2.5 \times 10^5$/24-well plate) harvested as indicated in the maturation protocol at day 6, were treated for 18 min with Control (DMSO), 2-Deoxy-Glucose (2-DG; 100 mM), Oligomycin (O; 1 μM), Etomoxir (4 μM) (Selleckchem, S8244), CB-839 (3 μM) (Selleckchem, S7655), a combination of 2DG and Oligomycin (DGO) or Harringtonine (H; 2 μg/mL). Following metabolic inhibitors, Puromycin (final concentration 10 μg/mL) was added to cultures for 17 min. After puromycin treatment, cells were detached from wells using TypLE Select (Fisher Scientific, 505914419), washed in cold PBS and stained with a combination of Human TureStain FcX (Biolegend, 422301) and fluorescent cell viability dye (Biolegend, 423105) for 10 min 4 °C in PBS. Following PBS wash step, primary antibodies

against surface markers were incubated for 25 min at 4 °C in Brilliant Stain Buffer (BD Biosciences, 563794). Cells were fixed and permeabilized using True-Nuclear Transcription Factor Buffer Set (Biolegend, 424401) as per manufacturer instructions. Intracellular staining of puromycin and protein targets was performed for 1 h in diluted (10x) permeabilization buffer at 4 °C. Finally, data acquisition was performed using the Cytek Aurora flow cytometer. Primary conjugated antibody information used in SCENITH panels are listed in Supplementary Tables 1 and 3. All antibodies were titrated to reduce spillover and increase resolution using single stained DC (generated as described above) samples. Unstained cell controls used for autofluorescence extraction were generated with additions of respective metabolic inhibitors (C, 2DG, O, DGO). Samples were unmixed using reference controls generated in combination with stained Ultracomp beads (Fisher Scientific, 01-2222-41) and stained cells using the SpectroFlo Software v2.2.0.1. The unmixed FCS files were used for data processing and analysis using CellEngine (CellCarta). For in-vitro cultured mDC, manually gated CD14⁻HLA-DR⁺CD86⁺ cells were used for downstream analysis. For median expression analyses MFI expression values from respective mDC and circulating cell populations from CellEngine were imported into R environment for correlation and heatmap clustering analyses using the below described R packages.

Calculations used to derive SCENITH parameters:

C = MFI of anti-Puro-Fluorochrome upon Control treatment
2DG = MFI of anti-Puro-Fluorochrome upon 2DG treatment
O = MFI of anti-Puro-Fluorochrome upon Oligomycin treatment
Eto = MFI of anti-Puro-Fluorochrome upon Etomoxir treatment
Tele = MFI of anti-Puro-Fluorochrome upon CB-839 treatment
DGO = MFI of anti-Puro-Fluorochrome upon 2DG + Oligomycin (DGO) treatment

Glucose dependence = $100(C - 2DG)/(C\text{-}DGO)$
Mitochondrial dependence = $100(C - O)/(C\text{-}DGO)$
FAO dependence = $100(C - Eto)/(C\text{-}DGO)$
Glutaminolysis dependence = $100(C - Tele)/(C\text{-}DGO)$
Glycolytic Capacity = $100 -$ Mitochondrial dependence
FAAO = $100 -$ Glucose dependence

## Single-cell metabolic regulome profiling (scMEP) by mass cytometry

scMEP analysis was performed as recently described. In short, monocytes and DC cultures were plated ($2.5 \times 10^6$/6-well plate) and harvested at desired time points. Antibodies targeting metabolic features were conjugated in-house using an optimized conjugation protocol[77] and validated on multiple sample types. Cells were prepared for scMEP analysis by incubation with small molecules to be able to assess biosynthesis rates of DNA, RNA and protein, cisplatin-based live/dead staining, PFA-based cell fixation and cryopreservation (dx.doi.org/10.17504/protocols.io.bkwkkxcw). Next, cells were stained with metabolic antibodies in a procedure that includes surface staining for 30 min at RT, PFA-fixation for 10 min at RT, MeOH-based permeabilization for 10 min on ice, intracellular staining for 1 h at RT and DNA intercalation (dx.doi.org/10.17504/protocols.io.bntnmeme). Finally, cells were acquired on a CyTOF2 mass cytometer (Fluidigm). Protein targets and antibody information used in scMEP are listed in Supplementary Table 2.

## Mass cytometry and spectral flow cytometry data processing and analysis

Raw mass spectrometry data were pre-processed, de-barcoded and imported into R environment using the flowCore package (version 2.0.1)[78]. Values were arcsinh transformed (cofactor 5) and normalized[30] for downstream analyses based on previously reported workflow[79]. Dimensionality reduction principal component analysis (PCA) and T-distributed stochastic neighbor embedding (tSNE) analyses were performed using stats (version 4.1.3) and Rtsne (version 0.15), respectively. Uniform Manifold Approximation and Projection (UMAP)

was performed using R package umap. (version 2.9.0). For visualization and heatmap clustering we utilized R packages ggplot2 (version 3.3.3) and ComplexHeatmap (version 2.4.3)[80], respectively. Stats (version 4.1.3) was used for linear regression analyses and Spearman correlation coefficient correlation matrix for marker expression profiles was computed and visualized using the corrr (version 0.4.3), Hmisc (version 4.5.0) and corrplot (version 0.88) R packages.

## Extracellular glucose and lactate measurements

Glucose and lactate levels were analyzed in DC culture supernatants using the BG1000 Blood Glucose Meter & test strips (Clarity, 75840-796) and Blood Lactate Measuring Meter Version 2 test strips (Nova Biomedical, Lactate Plus), respectively.

## Serum Luminex

The human immune monitoring 65-Plex (Thermo-Fisher Procarta Plex) was used to analyze pro-inflammatory cytokines in cell-free supernatants harvested from HD ($n = 4$) vs. melanoma patient ($n = 23$) mDC. The human Checkpoint 14-plex kit (Thermo-Fisher Procarta Plex) was also used for detection of culture supernatant checkpoint and costimulatory molecules.

## IFN-γ ELISPOT assays

Detailed methodology for melanoma antigen (MA)-specific T-cell responses is descried in refs. 40,41. To quantify specific responses to the melanoma antigens a positive response call (Yes) was defined as >10 spots counted per well and at least a twofold increase over baseline. To account for background, AdVLacZ response was subtracted from the AdV-melanoma antigen response.

## Statistical analysis

Clinical outcomes for analysis were described in detail previously, briefly: "good" outcomes were PR + SD > 6 mo.+ non-recurrent NED that was high risk at study entry (or NED1); "bad" outcomes were PD + SD ≤ 6 mo. + recurrent high risk NED[34,40,41].

Multi-group comparisons were tested by one-way ANOVA with Tukey's *post-hoc* test.

The Shapiro–Wilk test was used to asses data normality, and statistical tests were performed using R (version 3.6.1). Two-tailed Wilcoxon signed-rank test (non-normal data) and Two-tailed Student's t-test (normal data) was used for statistical analysis between 2 groups. Figure graphs were generated using the R package ggplot2 (version 3.1.1). Kaplan–Meier survival curve analysis and Cox proportional-hazards modeling were carried out using the R packages survival (version 3.1-8) and survminer (version 0.4.6).

**Study approval.** PBMCs from healthy donors were purchased (Trima Residuals RE202, Vitalant) which did not require an IRB approval. The clinical trial registration NCT01622933 and ethical approvals have been reported (ref. 13).

## Reporting summary

Further information on research design is available in the Nature Portfolio Reporting Summary linked to this article.

# Data availability

Data files and workflows generated for these analyses are available in a GitHub repository at https://github.com/ButterfieldLab/09-021-DC-Metabolism-Manuscript. The healthy donor microarray publicly available data used in this study are available in the GEO database under accession code GSE111581. The melanoma patient microarray data generated in this study have been deposited in the GEO database under accession code GSE157738. The remaining data are available within the article, Supplementary Information or Source Data file. Source data are provided with this paper.

## Code availability

Code and workflows generated for these analyses are available in a GitHub repository at [https://github.com/ButterfieldLab/09-021-DC-Metabolism-Manuscript].

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

## Acknowledgements

We acknowledge the financial support from the Parker Institute for Cancer Immunotherapy. We acknowledge the UCSF Parnassus Flow Cytometry Core Facility (RRID:SCR_018206) supported in part by Grant NIH P30 DK063720 and by the NIH S10 Instrumentation Grant S10 1S10OD026940-01. We thank both Ashley Menk and Dr. Greg Delgoffe for assistance with the Seahorse assays (Univ. Pittsburgh). We also acknowledge ANR and Inserm Transfert for the ANR-20-CE-CE14-0028-01 and CoPoC MAT-PI-17493-A-04 grants to R.J.A.

## Author contributions

Study design: J.A. and L.H.B.; Study conduct: J.A., P.V.M., D.M.M., F.J.H.; Data interpretation: J.A., P.V.M., L.H.B.; Key reagents, advice and specimens: F.J.H., R.J.A., S.C.B.; Drafted the manuscript: J.A., L.H.B.; Revised the manuscript: J.A., P.V.M., R.J.A., L.H.B.; Reviewed, edited and approved the manuscript: J.A., P.V.M., D.M.M., F.J.H., S.C.B., R.J.A., L.H.B.

## Competing interests

J.A., P.V.M., D.M.M., F.J.H., S.C.B.: declare no competing interests. L.H.B. declares the following unrelated competing interests: Calidi Scientific and Medical Advisory Board, April 6, 2017–2023; KaliVir, Scientific

Advisory Board, 2018–2023; Torque Therapeutics, Scientific Advisory Board, 2018–2020; Khloris, Scientific Advisory Board, 2019–2023; Pyxis, Scientific Advisory Board, 2019–2023; CytomX, Scientific Advisory Board, 2019–2023; DCprime, Scientific Advisory Board meeting, 2020; RAPT, Scientific Advisory Board, 2020–2023; Takeda, Scientific Advisor, 2020–2023; EnaraBio scientific advisor, Feb. 2021. R.J.A. declares no competing interests. There are restrictions to the commercial use of SCENITH due to a pending patent application (PCT/EP2020/WO2020212362A1).
