## [Peer Review File · Nature Communications]

Immuno-Metabolic Dendritic Cell Vaccine Signatures Associate with Overall Survival in Vaccinated Melanoma PatientsREVIEWER COMMENTS

Reviewer #1 (Remarks to the Author): with expertise in DC, immuno-metabolism

Adamik and colleagues have characterized and compared in great detail the immunometabolic properties of moDCs used as cellular therapy to treat late-stage melanoma patients with those from healthy controls. Using a combination of techniques they convincingly show that moDC from melanoma patients are metabolically skewed towards glycolysis and display reduced oxphos and that this metabolic state is negatively associated with overall survival. Finally, they perform a similar analysis on the myeloid compartment in blood from patients and healthy controls and find largely a similar metabolic skewing with lower mitochondrial dependence. Although the work is well presented and novel, I have a couple of concerns and comments that I would like to see addressed before it would be suitable for publication in Nat Comms.

1) My main point of critique relates to the absence of a correlation between metabolic state and functional output of the moDCs and ex vivo DCs as the authors do not find any association with T cell responses. To me this suggests that, as the authors only at the very end allude to, the metabolic differences between patient and HD DCs are more a reflection of the diseased state rather than that they underpin functional properties of these DCs. To evaluate whether there is a direct mechanistic link between the described metabolic shift and immunophenotype the authors should evaluate whether inhibition of mitochondrial metabolism and/or promoting glycolysis in HD moDCs leads to changes in expression of markers that resemble those of melanoma moDCs.

2) Related to this, is there any metabolic parameter in DCs that correlates with the Luminex data that are presented in Fig 6? Now this dataset is a bit disconnected from the rest.

3) How well does the metabolic state of monocytes as determined in fig 7, predict the metabolic state of the moDCs that are cultured from these monocytes. This would be

4) Please better explain in the introduction or at the start of the results section how mDCs used as immune cell therapy are generated (derived from monocytes etc). If you are not familiar with this field, then this is currently not clear.

5) The authors show that the more glycolytic populations are characterized by higher CD86, CD40 and HLADR as well as some regulatory molecules. To me this suggests an overall more immunogenic phenotype. Yet this metabolic state is associated with poorer disease

outcome. The authors should provide their view on this apparent discrepancy.

6) Y-axes in various figure panels seem not to be correctly labelled. For instance in Fig 2 OCR and ECAR levels are very very low if this is true. I imagine the data represent some normalized value. This should be clarified.

7) As HD only 3 individuals are included in most figures. As a result in many cases comparisons between HD and patients DCs are borderline significant. To strengthen the data and conclusions, HD group should be increased in number.

8) In the discussion the authors start by stating 'multiple platforms were used to identify mechanisms of DC function....'. I do not think the authors are able to make that statement as all data are correlative. See also point 1. So this should be rephrased.

9) In the discussion the authors state that 'we speculate that the aberrant increase in glycolysis may reflect a transitional state with immune maturation delay, which resemble the highly glycolytic phenotype of maturation deficient tolDC.' It is unclear how the authors arrive at this idea, as they show that the more glycolytic populations are characterized by higher expression of maturation markers, the exact opposite from the maturation deficient tolDC, the authors compare them too.

10) The authors are encouraged to discuss how they envision these findings could help to improve DC-based vaccine efficacy, and not only serve as a predictor of success of a therapy.

11) Please show the frequencies of the myeloid cell populations in a main figure (% of PBMCs). These data are now buried in supFig5 and only shown as frequencies of parent populations, rather than Frequencies within total PBMCs

Reviewer #2 (Remarks to the Author): with expertise in melanoma, DC-based cancer vaccines

The manuscript pioneers by providing a broad and detailed overview of the metabolic profile of DCs in both a healthy and disease (cancer) setting, using both established (Seahorse) and more novel, innovative techniques (SCENITH, scMEP). The link to both the phenotypic, functional and transcriptional level is of added value. However, the paper is highly descriptive and could benefit from highlighting the most remarkable results, how these are confirmed by the distinct profiling methods, and intermediate conclusions. Moreover, some in depth analysis about the underlying mechanisms of some of these

findings (f.e. mitochondrial dependency) could further support the results. In addition, for readers less experienced in metabolism, the manuscript would benefit from explaining the technologies used and what is represented by the outcome measures of these assays.

Major comments:

- It would be good to add actual 'timepoints' to the schematic representation in Fig1A indicating the stages of profiling to link them to the methodology described in materials & methods.
- Transcriptional analysis focused on mDCs as most transcriptional changes occurred upon DC maturation (lines 147-150). Could the authors comment on the number of DEGs in adenovirally engineered DCs in comparison to mDCs and describe pathways related to these DEGs?
- The authors use healthy donors as comparison to melanoma DC. Metabolic dysfunctions can be age-related. Are the healthy donors age-matched with patients?
- mDC maturation has been performed using LPS and IFN γ . How would the data presented in this manuscript be affected upon usage of distinct maturation stimuli?
- It would be more correct to indicate that the 725 upregulated and 818 (line 151) downregulated genes were specific to the melanoma mDCs.
- If one would start from the upregulated genes specific for melanoma mDCs (Fig 1B; 725 genes) and compare these between bad and good outcome, could this reveal potential biomarkers that predict the response of the patient to therapy?
- Figure 1D provides information concerning different pathways between good and bad outcome groups in mDCs. These responses are based on the therapy used in the clinical trial (Butterfield et al, 2019). How translatable would these results be to other therapies?
- Some explanation about the metabolic parameters acquired by Seahorse analysis (lines 179-183) would help the understanding for non-metabolite experts.
- Does the mitochondrial dysfunction claimed in line 188-191 in melanoma mDCs relate to changes in mitochondrial biogenesis?
- It was shown by the authors that enhanced metabolic dependency on mitochondrial metabolism, FAO and glutaminolysis correlated with positive outcome. However, these dependencies do not (at least not significantly) associate with an Ag-specific T cell response (lines 217-220). Could the authors comment on which other mechanism(s), driven by mDC

metabolism, could explain the effects on outcome?

- The authors claim that p-mTOR and p-AMPK levels tend to increase in bad outcome mDCs (line 236). However, these data are not significant and the trend is not convincing.
- Given the combination of markers (especially ILT3 and PDL1) that associate with the most glycolytic mDCs, the term 'inflammatory surface markers' (line 274 + 223) is not well-chosen.
- The authors demonstrate increased expression of MCT1 in melanoma mDCs and correlate it to lactate levels. However, other monocarboxylate transporters can be involved in lactate secretion/uptake. Addition of MCT2/3/4 expression data would be required.
- Supp Figure 4S shows the correlation between glucose uptake and lactate secretion. However, it is likely that (especially in mDCs where this correlation is less pronounced) part of the glucose is fueled elsewhere. Calculating the fraction of glucose that is being converted to lactate, upon calculation of the lactate secretion/glucose uptake ratio could provide this information.
- Based on Seahorse, SCENITH and scMEP data, mitochondrial dependence in mDCs is highlighted as an interesting biomarker of clinical response in the discussion. Could the authors comment on a specific (set of) biomarkers one could screen for on protein level that is feasible to implement in clinical trials? Would it be possible to screen for these biomarkers on a transcriptional level as well?

Textual changes:

- Line 2: spelling mistake: OCAR instead of OCR.
- Line 84-86: sentence structure seems incorrect.
- Line 111: bracket comes before abbreviation 'CyTOF'.
- Line 150: introduction of the 'HD' (healthy donor) abbreviation is missing.
- Line 185: there is no reference in the text to Figure 2C.
- Line 193: Increase should be increased
- Line 201: one should refer to Supp Figure 2A instead of 1A.
- Line 289: the order of the 'good' and 'bad outcome groups' is different in the text and Figure (5B).
- The reference to Supp Fig 5A on line 345 is incorrect.
- Line 428: While in the inhibitory effects... should be While the inhibitory effects...

- Line 429: was described should be were described

Reviewer #3 (Remarks to the Author): with expertise in immuno-metabolism

In their manuscript „Immuno-metabolic dendritic cell vaccine signatures associate with overall survival in vaccinated melanoma patients“, Adamik et al. analyzed the transcriptomic and immune-metabolic profile of monocyte-derived DC from late-stage melanoma patients. They first compared immature and mature DCs from melanoma patients with publicly available microarray data on healthy donors. They further used different methods (Seahorse, SCENITH, scMEP) to describe the metabolic profile of „mDC“ from healthy donors and melanoma patients. My major concern about this publication is that is not possible to evaluate the meaning of the results due to several flaws:

- The description of the patients used in the study is incomplete: there is no background information on how this study was carried on, not even an ethical vote. For some experiments, the patients are separated according to their outcome based on the criteria GOOD as „PR+SD>6 mo. + non-recurrent NED“ or BAD as „PD+SD</=6 mo. + recurrent NED“. Later the samples are distinguished based on PR, SD, NED1, NED2, or PD without a clear definition of these parameters.
- The comparisons using data from HD DC taken from a repository that was generated using a different protocol (3d vs. 5d culture) are questionable. How can the authors be sure that all the differences observed in gene expression are due to the melanoma and not to the culture conditions?
- The M&M section, the figure description, and the description of the results are not sound. There is no clear description of the protocol used to obtain the cells and any other information required to evaluate the validity of the results is missing. For example: in M&M they described „melanoma patients elutriated fraction 5 cells, HD PMBC and monocytic mDC cultures“. But in the description of the figures they used mDC or melanoma iDC and mDC, mDC culture supernatants, circulating myeloid/DC subtype populations. The source of the "DC" and the way they were obtained are crucial parameters to interpret the data.
- The apparent time points indicated in Fig.1A are not really indicated in the figure
- Several figures refer to the HD group but there is no data for HD

Altogether the manuscript gives a very sloppy impression (e.g. including many typos) and the description of the results does not follow any logic. It seems that the authors do not know why they made the experiments.

Reviewer #4 (Remarks to the Author): with expertise in metabolism

Adamik et al. present results for interrogating the metabolism of dendritic cells and other related cell types from melanoma patients and healthy donors. They use multiple technologies including microarrays, Seahorse, SCENITH, scMEP, and cell type profiling using markers.

The manuscript focuses on the very challenging topic to find the markers of clinical outcomes.

The results are extensive and represent a great and extensive resource characterizing the metabolism of DCs from healthy donors and melanoma patients.

My key concern is about the claims about the clinical outcomes. Most if not all results do not show any significant differences between “bad” and “good” clinical outcomes. Most (if not all) claims about the outcomes are based on “trends” namely changes of average values, however, all p-values are greater than 0.05. So, I strongly propose the authors to remove the statements and claims about being able to discern the clinical outcomes as I believe they have no significant evidence for it.

However, the manuscript is still worth publishing in Nature Communications as it contains important results about metabolic differences between healthy donors and melanoma patients.

Another point I would like to bring up is the interpretation of oligomycin-treated cells. The authors interpret the lack of change of the protein translation after the oligomycin treatment as cells being in the glycolytic state. Can authors prove it somehow? Although I see that the lack of the change indicates that the cells are not in the mitochondrial state,

can it be that the cells rely on another pathway than glycolysis, e.g. PPP?

Finally, the authors should define all abbreviations including HD, iDC, OS, PFS, and many more.

Point by Point Response

Reviewer's Comments:

Reviewer #1 (Remarks to the Author)

Adamik and colleagues have characterized and compared in great detail the immunometabolic properties of moDCs used as cellular therapy to treat late-stage melanoma patients with those from healthy controls. Using a combination of techniques they convincingly show that moDC from melanoma patients are metabolically skewed towards glycolysis and display reduced oxphos and that this metabolic state is negatively associated with overall survival. Finally, they perform a similar analysis on the myeloid compartment in blood from patients and healthy controls and find largely a similar metabolic skewing with lower mitochondrial dependence.

Response to Reviewer 1: we thank the reviewer for these comments.

Although the work is well presented and novel, I have a couple of concerns and comments that I would like to see addressed before it would be suitable for publication in Nat Comms.

1) My main point of critique relates to the absence of a correlation between metabolic state and functional output of the moDCs and ex vivo DCs as the authors do not find any association with T cell responses. To me this suggests that, as the authors only at the very end allude to, the metabolic differences between patient and HD DCs are more a reflection of the diseased state rather than that they underpin functional properties of these DCs. To evaluate whether there is a direct mechanistic link between the described metabolic shift and immunophenotype the authors should evaluate whether inhibition of mitochondrial metabolism and/or promoting glycolysis in HD moDCs leads to changes in expression of markers that resemble those of melanoma moDCs.

Response to Reviewer 1: We agree with the point raised by the reviewer. The lack of significance of the correlations between immune response and DC vaccine metabolism (shown in detail throughout Suppl. Figure 3), was unexpected. To respond to the reviewer's specific question tying mitochondrial metabolism to DC phenotype, we have performed new studies.

Specifically, to demonstrate the functional impact of alteration of mitochondrial metabolism, we have conducted a new experiment to inhibit p-AMPK, a well-established positive regulator of mitochondrial metabolism. Increased concentrations of the inhibitor Dorsomorphin inhibited phosphorylation of AMPK in a dose-dependent manner and resulted in reduced expression of several important DC cell surface proteins including HLA-DR, CD86, PD-L1 and CD206 in HD mDC. These data are now shown in new Supplemental Figure 4A and described in the text (Lines 263:268).

2) Related to this, is there any metabolic parameter in DCs that correlates with the Luminex data that are presented in Fig 6? Now this dataset is a bit disconnected from the rest.

Response to Reviewer 1: To address this, we have now performed additional correlation analyses that are presented in figure 6C. The text has been edited to describe these data in lines 378:382, describing new observations with glycolysis and OxPhos and DC secreted proteins.

3) How well does the metabolic state of monocytes as determined in fig 7, predict the metabolic state of the moDCs that are cultured from these monocytes.

Response to Reviewer 1: This is an important question. To address this, we have added new Suppl. Figure 7C-E, showing the ability of the metabolic state of monocyte subsets to predict the mDC state. Linear regression was used to test whether the metabolic state of circulating monocyte populations can predict metabolism of cultured moDCs. We show that intermediate monocyte (iMo) state was the most significant predictor of mDC state. Classical monocytes' (cMo) glycolytic capacity and mitochondrial dependence also showed trends (0.056, 0.092) for mDC state. These results are now described in the results section of the manuscript in lines 438:442

4) Please better explain in the introduction or at the start of the results section how mDCs used as immune cell therapy are generated (derived from monocytes etc). If you are not familiar with this field, then this is currently not clear.

Response to Reviewer 1: We apologize for the limited reference to the published clinical trial. We agree that some additional detail is needed and have added to the Results section, Methods section and Figure 1.

5) The authors show that the more glycolytic populations are characterized by higher CD86, CD40 and HLADR as well as some regulatory molecules. To me this suggests an overall more immunogenic phenotype. Yet this metabolic state is associated with poorer disease outcome. The authors should provide their view on this apparent discrepancy.

Response to Reviewer 1: This is an interesting point. While those phenotypes haven't traditionally correlated with immune responses or clinical outcomes (as we noted in the clinical trial report, reference #40), indeed, the higher and more variable expression of these immune markers present on the glycolytic cell populations are enriched in the worse outcome group samples. Direct comparisons with the high mitochondrial metabolism cells do not show this discrepancy. In fact, we see no change or downregulation of CD86, CD40 and PD-L1. The discrepancy comes from the cell populations within a given well that are glycolytic and exhibit increased DC marker expression. This profile does not reflect mitochondrially inhibited or tolerogenic DC phenotype.

The data points to a chronic glycolytic metabolic state in the melanoma mDCs that has been recently described in study by Maio et al. (ref. #64), which showed that a preexisting pathologically active glycolytic state termed "exhausted glycolysis" in monocytes from tuberculosis patients limited the generation and migratory capacities of monocyte-derived DCs with deregulated expression of surface markers, including CD83 and CD86. We have further addressed this in the Discussion.

6) Y-axes in various figure panels seem not to be correctly labelled. For instance in Fig 2 OCR and ECAR levels are very very low if this is true. I imagine the data represent some normalized value. This should be clarified.

Response to Reviewer 1: Thank you for pointing this out, we have scaled the OCR pmol/min and ECAR mpH/min values to 0 to 1 range by subtracting the minimum and dividing by the maximum values for cleaner plotting. This allowed for more uniform values for additional comparisons and correlations in some downstream analyses. We have adjusted the labels and description in the legends.

7) As HD only 3 individuals are included in most figures. As a results in many cases comparisons between HD and patients DCs are borderline significant. The strengthen the data and conclusions, HD group should be increased in number.

Response to Reviewer 1: We appreciate the question. All HD and patient samples were cultured, stained and flow cytometrically analyzed on the same day to prevent batch effects coming from both technical and biological variations (particularly using the Cytex instrument). For some of the analyses we were able to include 4 HD, but the large scale nature of these experiments allowed for 3 HD samples, which generally showed much less variation than patient cells.

8) In the discussion the authors start by stating 'multiple platforms were used to identify mechanisms of DC function....'. I do not think the authors are able to make that statement as all data are correlative. See also point 1. So this should be rephrased.

Response to Reviewer 1: We have rephrased the sentence and have also added new mitochondrial metabolism signaling inhibition studies.

9) In the discussion the authors state that 'we speculate that the aberrant increase in glycolysis may reflect a transitional state with immune maturation delay, which resemble the highly glycolytic phenotype of maturation deficient toIDC.' It is unclear how the authors arrive at this idea, as they show that the more glycolytic populations are characterized by higher expression of maturation markers, the exact opposite from the maturation deficient toIDC, the authors compare them too.

Response to Reviewer 1: We now address this more fully in the Discussion. Please also see response #5 above.

10) The authors are encouraged to discuss how they envision these findings could help to improve DC-based vaccine efficacy, and not only serve as a predictor of success of a therapy.

Response to Reviewer 1: We have identified dysfunctional states of circulating monocytes and DC which may impact vaccines which deliver antigen and rely on endogenous antigen presenting cells. We suggest that metabolism-skewing combination approaches to alter this state as a future direction. Ex vivo DC culture conditions need to be optimized to avoid the primary reliance on glycolysis in cancer patient-derived cells for optimal outcomes. We have added these ideas to the Discussion.

11) Please show the frequencies of the myeloid cell populations in a main figure (% of PBMCs). These data are now buried in supFig5 and only shown as frequencies of parent populations, rather than Frequencies within total PBMCs

Response to Reviewer 1: We agree with the reviewer that these data would be interesting. Because most of these cells were obtained from patient leukapheresis cells which were elutriated into myeloid and lymphoid fractions, and there were very few (if any) T, B, NK cells in these myeloid fractions, hence the numbers would not be relatable to whole PBMC gating. We were, however, able to calculate the population frequencies relative to Non-NK cells populations. We have generated a table (added to Suppl. Figure 6) which shows the calculated myeloid subset frequencies.

Reviewer #2 (Remarks to the Author)

The manuscript pioneers by providing a broad and detailed overview of the metabolic profile of DCs in both a healthy and disease (cancer) setting, using both established (Seahorse) and more novel, innovative techniques (SCENITH, scMEP). The link to both the phenotypic, functional and transcriptional level is of added value. However, the paper is highly descriptive and could benefit from highlighting the most remarkable results, how these are confirmed by the distinct profiling methods, and intermediate conclusions.

Moreover, some in depth analysis about the underlying mechanisms of some of these findings (f.e. mitochondrial dependency) could further support the results.

In addition, for readers less experienced in metabolism, the manuscript would benefit from explaining the technologies used and what is represented by the outcome measures of these assays.

Response to Reviewer 2:

We appreciate these comments and have more clearly emphasized some key conclusions in the revised Discussion.

The revised manuscript adds new mechanistic data on mitochondrial dependence signaling mechanisms and new data analysis that sheds further light on the work.

We appreciate these comments and have better described the different metabolic assays in the revised text.

Major comments:

- It would be good to add actual 'timepoints' to the schematic representation in Fig1A indicating the stages of profiling to link them to the methodology described in materials & methods.

Response to Reviewer 2: We have adjusted this figure to reflect the time points associated with each maturation stage and experimental approach.

- Transcriptional analysis focused on mDCs as most transcriptional changes occurred upon DC maturation (lines 147-150). Could the authors comment on the number of DEGs in adenovirally engineered DCs in comparison to mDCs and describe pathways related to these DEGs?

Response to Reviewer 2: We have performed additional differential expression and pathway enrichment analysis to illustrate the changes associated with virally engineered (post-maturation) DC shown in Supplemental figure 1B and described the results in lines: 147-149, noting that the AdV-engineering step leads to changes in 82 genes.

- The authors use healthy donors as comparison to melanoma DC. Metabolic dysfunctions can be age-related. Are the healthy donors age-matched with patients?

Response to Reviewer 2: Because healthy donor cells were purchased from a vendor, we did not have access to the age range of the donors for this study.

- mDC maturation has been performed using LPS and IFN γ . How would the data presented in this manuscript be affected upon usage of distinct maturation stimuli?

Response to Reviewer 2: We hypothesize that the maturation stimulus would significantly change the metabolic profile, with some commonality between different Th1-skewing approaches and Th2 or regulatory triggers related to DC maturation in general. To address this experimentally, we recently published two reports which answer this question with healthy donor DC.

In Adamik 2022 (ref. #17) we find that Vitamin D3-cultured tolerogenic DC are skewed towards glycolysis, which is similar to DC exposed to the HCC tumor protein alpha fetoprotein which become glycolytic (Munson 2023, ref. #68). These publications add additional DC activation or maturation triggers which alter their metabolic profile.

- It would be more correct to indicate that the 725 upregulated and 818 (line 151) downregulated genes were specific to the melanoma mDCs.

Response to Reviewer 2: We have revised the sentence in lines 152-154.

- If one would start from the upregulated genes specific for melanoma mDCS (Fig 1B; 725 genes) and compare these between bad and good outcome, could this reveal potential biomarkers that predict the response of the patient to therapy?

Response to Reviewer 2: We thank the reviewer for this suggestion. We have further evaluated 57 genes differentially expressed in the mDCs between good and bad outcome groups. We have added Supplemental Figure 1C as hierarchical clustering heatmap represented these upregulated genes with unsupervised clustering of the good and bad outcome groups. We have applied this gene signature in a gene set variation analysis to further evaluate its potential to separate Good and Bad outcome mDC. This is described in the text in lines (171 - 178) with a suggested need to validate such a signature prospectively.

- Figure 1D provides information concerning different pathways between good and bad outcome groups in mDCs. These responses are based on the therapy used in the clinical trial (Butterfield et al, 2019). How translatable would these results be to other therapies?

Response to Reviewer 2: We appreciate this question. Many of the differentially regulated genes are involved in DC maturation (and not specific to the adenovirus we used to engineer the DC, nor the melanoma-associated antigens we loaded the DC with). There is an obvious relevance to DC based vaccines (which have been tested in over 200 clinical trials). We have seen related metabolic trends in our two recent studies with tolerogenic DC (unrelated to melanoma (ref. #17) and AFP-exposed DC (related to HCC, Ref. #68), hence, these data also point to DC metabolic themes that have relevance for broader settings than a single melanoma cancer vaccine trial. The circulating monocyte and DC data are relevant to antigen-delivery based vaccines, which we now suggest in the Discussion.

- Some explanation about the metabolic parameters acquired by Seahorse analysis (lines 179-183) would help the understanding for non-metabolite experts.

Response to Reviewer 2: We have more clearly introduced the Seahorse assay in this revised manuscript.

- Does the mitochondrial dysfunction claimed in line 188-191 in melanoma mDCs relate to changes in mitochondrial biogenesis?

Response to Reviewer 2: We appreciate this question. To address thus, using scMEP profiling we have included new data for PGC1alpha for monitoring changes in mitochondrial biogenesis and function. Consistent with increased proton leak across the cell membrane as measured by the Seahorse assay (Figure 2D), PGC1α was downregulated in melanoma DC suggesting their impaired biogenesis may contribute to the observed metabolic dysfunction in the melanoma patients (new data in Suppl. Fig. 5A, top-right). We have modified the corresponding corresponding text in lines 303:308, pg. 10-11.

- It was shown by the authors that enhanced metabolic dependency on mitochondrial metabolism, FAO and glutaminolysis correlated with positive outcome. However, these dependencies do not (at least not significantly) associate with an Ag-specific T cell response (lines 217-220). Could the authors comment on which other mechanism(s), driven by mDC metabolism, could explain the effects on outcome?

Response to Reviewer 2: We agree with the important point raised by the reviewer and comment as follows:

One reason for this observation may be that during our profiling we can mainly capture the metabolism of DC at steady state. This may not be reflective of the metabolic changes occurring during antigen-transduction and T cell priming. Evidence now suggest that DC utilize different metabolic states to drive polarization of different Th cell subsets. Because T cell response activation is a highly dynamic and context-dependent process, the steady state metabolic profiling may be upstream of critical subsequent metabolic changes more predictive for T cell priming responses.

We also hypothesize that metabolism may affect other aspects of DC biology including survival and migratory capacity to lymph nodes that plays large role on their efficacy to mount successful immune response in patients. While we did measure antigen-specific responses in most patients, the frequency of these was quite variable and therefore challenging to correlate with our metabolic profiling. Our data may also point out to a chronic glycolytic metabolic state in the melanoma mDCs that may represent a cancer-induced chronic metabolic impairment similar to one described by Maio et al. These authors showed a presence of preexisting pathologically active glycolytic state termed “exhausted glycolysis” in monocytes from tuberculosis patients, which limited generation and migratory capacities of monocyte-derived DCs. We added this reasoning to the Discussion, lines 517:523.

- The authors claim that p-mTOR and p-AMPK levels tend to increase in bad outcome mDCs (line 236). However, these data are not significant and the trend is not convincing.

Response to Reviewer 2: We understand the concerns raised by the reviewer and therefore we have removed those particular data and re-focused the pAMPK analysis to more significant changes and impact (in Figure 4C) between the mitochondrial and glycolytic mDC groups. In addition, to strengthen our results regarding the functional impact of p-AMPK and mitochondrial metabolism alteration, we have now conducted a new experiment to inhibit p-AMPK (a well-established positive regulator of mitochondrial metabolism). Increased concentrations of the Dorsomorphin inhibitor, pAMPK concentration dependent decrease in AMPK phosphorylation and resulted in reduced expression of several immune makers including HLA-DR, CD86, PD-L1 and CD206 in HD mDC. These data are represented in Supplemental Figure 4, and text lines 263:268.

- Given the combination of markers (especially ILT3 and PDL1) that associate with the most glycolytic mDCs, the term ‘inflammatory surface markers’ (line 274 + 223) is not well-chosen.

Response to Reviewer 2: We agree with the reviewer and we have restated the statement and focused more on the fact that these glycolytic cell populations seem less uniform in terms of their immune profiles. Collectively, these results suggest that mDC from melanoma patients, and, predominantly of the worst outcome group, contain highest number of glycolytic cells, which represent less uniform cell populations and exhibit variable expression of multiple immune surface markers. See text lines 282:285.

- The authors demonstrate increased expression of MCT1 in melanoma mDCs and correlate it to lactate levels. However, other monocarboxylate transporters can be involved in lactate secretion/uptake. Addition of MCT2/3/4 expression data would be required.

Response to Reviewer 2: We appreciate this question. While we do not have protein-level data for additional MCT transporter family receptors, we note that in our recent report (Adamik 2022 ref. #17) we tested functional inhibition of MCT1 to show that it significantly reduced lactate in culture media. In the present work, we examined gene expression profiles for MCT2 (SLC16A7), MCT3 (SLC16A8) and MCT4 (SLC16A4), and did not observe significant correlations between the additional MCT family transporter expression levels and lactate secretion at either the iDC or mDC stages (new analysis shown in Supp Figure 4D). In addition, we did observe that MCT1 is the highest expressed transporter in mDC (Supp Figure 4E), which further supports that its expression may reflect its importance in the context of moDC biology. We added these new observations in the Results section in lines 340:342.

- Supp Figure 4S shows the correlation between glucose uptake and lactate secretion. However, it is likely that (especially in mDCs where this correlation is less pronounced) part of the glucose is fueled elsewhere. Calculating the fraction of glucose that is being converted to lactate, upon calculation of the lactate secretion/glucose uptake ratio could provide this information.

Response to Reviewer 2: This is an interesting question. To address this, and, based on the suggested formula, we have calculated the fraction of glucose that is being converted to lactate in the HD, Good, and Bad patient outcome groups in both iDC and mDC. We observe a significant increase of lactate being converted to glucose in the melanoma mDCs as compared to HD. No significant difference between the two outcome groups is observed. These results are described in lines: 338:340.

- Based on Seahorse, SCENITH and scMEP data, mitochondrial dependence in mDCs is highlighted as an interesting biomarker of clinical response in the discussion. Could the authors comment on a specific (set of) biomarkers one could screen for on protein level that is feasible to implement in clinical trials? Would it be possible to screen for these biomarkers on a transcriptional level as well?

Response to Reviewer 2: To address this suggestion, we have added more specific suggested biomarkers of DC vaccines in the Discussion. These remain to be tested prospectively, but RNA and protein level targets more quantitative, and lactate secretion and mitochondrial dependence may serve as the starting place for this.

Textual changes:

- Line 2: spelling mistake: OCAR instead of OCR.
- Line 84-86: sentence structure seems incorrect.
- Line 111: bracket comes before abbreviation 'CyTOF'.
- Line 150: introduction of the 'HD' (healthy donor) abbreviation is missing.
- Line 185: there is no reference in the text to Figure 2C.
- Line 193: Increase should be increased
- Line 201: one should refer to Supp Figure 2A instead of 1A.
- Line 289: the order of the 'good' and 'bad outcome groups' is different in the text and Figure (5B).
- The reference to Supp Fig 5A on line 345 is incorrect.
- Line 428: While in the inhibitory effects... should be While the inhibitory effects...
- Line 429: was described should be were described

These textual changes were corrected.

Reviewer #3 (Remarks to the Author): with expertise in immuno-metabolism

In their manuscript „Immuno-metabolic dendritic cell vaccine signatures associate with overall survival in vaccinated melanoma patients“, Adamik et al. analyzed the transcriptomic and immune-metabolic

profile of monocyte-derived DC from late-stage melanoma patients. They first compared immature and mature DCs from melanoma patients with publicly available microarray data on healthy donors. They further used different methods (Seahorse, SCENITH, scMEP) to describe the metabolic profile of „mDC“ from healthy donors and melanoma patients. My major concern about this publication is that it is not possible to evaluate the meaning of the results due to several flaws:

- The description of the patients used in the study is incomplete: there is no background information on how this study was carried on, not even an ethical vote. For some experiments, the patients are separated according to their outcome based on the criteria GOOD as „PR+SD>6 mo. + non-recurrent NED“ or BAD as „PD+SD</=6 mo. + recurrent NED“. Later the samples are distinguished based on PR, SD, NED1, NED2, or PD without a clear definition of these parameters.

Response to Reviewer 3: We apologize for our lack of clarity around these clinical details and trial analysis in this report, as we published the trial and referred to and cited it but attempted to minimize repeating text. We have now added details to the beginning of the Results and Methods sections. We defer to the Editors as to the extent to which we should repeat published information.

- The comparisons using data from HD DC taken from a repository that was generated using a different protocol (3d vs. 5d culture) are questionable. How can the authors be sure that all the differences observed in gene expression are due to the melanoma and not to the culture conditions?

Response to Reviewer 3: We acknowledge the concern raised. To respond, our microarray gene expression comparisons between HD and melanoma DC specifically focus on the mDC stage in which both protocols used the same maturation cocktail and 24h maturation time. Therefore, changes in the pathways are specifically reflecting maturation induced or lack thereof. Studies examining differences in immature iDC generation revealed that monocytes cultured in the presence of IL4+GM-CSF within 48 hours exhibit iDC characteristics, and upon maturation, these cells displayed a fully mature mDC immunophenotype (Dauer et al., J Immunol (2003) 170 (8): 4069–4076). Therefore, we do not expect the 3 vs 5 days of culture exhibit substantial differences in iDC generation. In addition, global gene expression comparisons of maturation of monocyte-derived DCs with 3 different cocktails LPS+IFN γ , LPS+IFN γ +IL1b and LPS+IFN γ +TNF α revealed only 13 differentially expressed genes among the 3 maturation groups suggesting that even the presence of additional maturation stimuli would not likely impact our HD vs melanoma mDC analysis (Han et al., J Immunother. 2009 May; 32(4): 399–407). We have also added a statement regarding this point in lines 152: 155.

- The M&M section, the figure description, and the description of the results are not sound. There is no clear description of the protocol used to obtain the cells and any other information required to evaluate the validity of the results is missing. For example: in M&M they described „melanoma patients elutriated fraction 5 cells, HD PMBC and monocytic mDC cultures“. But in the description of the figures they used mDC or melanoma iDC and mDC, mDC culture supernatants, circulating myeloid/DC subtype populations. The source of the "DC" and the way they were obtained are crucial parameters to interpret the data.

Response to Reviewer 3: We apologize for our lack of clarity, as we published the trial and referred to it but attempted to minimize repeating text. We have added additional information to the Methods and Results sections.

- The apparent time points indicated in Fig.1A are not really indicated in the figure

Response to Reviewer 3: We have adjusted this figure to reflect the time points associated with each maturation stage and experimental approach.

- Several figures refer to the HD group but there is no data for HD

Response to Reviewer 3: We apologize for this oversight and have adjusted the text to reflect use of HD data comparisons.

Altogether the manuscript gives a very sloppy impression (e.g. including many typos) and the description of the results does not follow any logic. It seems that the authors do not know why they made the experiments.

Response to Reviewer 3: We are sorry to read this broad criticism and hope that this revision clarified the reviewer's concerns and he/she finds the manuscript clear and logical.

Reviewer #4 (Remarks to the Author): with expertise in metabolism

Adamik et al. present results for interrogating the metabolism of dendritic cells and other related cell types from melanoma patients and healthy donors. They use multiple technologies including microarrays, Seahorse, SCENITH, scMEP, and cell type profiling using markers.

The manuscript focuses on the very challenging topic to find the markers of clinical outcomes.

Response to Reviewer 4: We appreciate this comment.

The results are extensive and represent a great and extensive resource characterizing the metabolism of DCs from healthy donors and melanoma patients.

Response to Reviewer 4: We appreciate this comment.

My key concern is about the claims about the clinical outcomes. Most if not all results do not show any significant differences between “bad” and “good” clinical outcomes. Most (if not all) claims about the outcomes are based on “trends” namely changes of average values, however, all p-values are greater than 0.05. So, I strongly propose the authors to remove the statements and claims about being able to discern the clinical outcomes as I believe they have no significant evidence for it.

Response to Reviewer 4: We agree that some of the data are correlative with clinical outcome and other data are not. We have presented many significant correlations as well as a number of statistical trends. We have revised our statements to ensure we are not over interpreting the differences between the outcome groups.

However, the manuscript is still worth publishing in Nature Communications as it contains important results about metabolic differences between healthy donors and melanoma patients.

Response to Reviewer 4: We appreciate this response.

Another point I would like to bring up is the interpretation of oligomycin-treated cells. The authors interpret the lack of change of the protein translation after the oligomycin treatment as cells being in the glycolytic state. Can authors prove it somehow? Although I see that the lack of the change indicates that the cells are not in the mitochondrial state, can it be that the cells rely on another pathway than

glycolysis, e.g. PPP?

Response to Reviewer 4: Our interpretation of oligomycin-treated cells not demonstrating a change in protein translation is based on the principle that oligomycin inhibits mitochondrial ATP production, leading cells to depend on glycolysis for their energy needs. When ATP production is inhibited in the mitochondria, cells typically upregulate glycolysis to compensate for the energy deficit.

Regarding the PPP, this pathway primarily functions in the generation of NADPH and ribose 5-phosphate, which are critical for cellular biosynthetic processes, such as nucleotide and lipid synthesis, but it is not a major contributor to ATP production. In fact, the PPP is an oxidative process and does not lead to the production of ATP. Therefore, under conditions of oligomycin treatment, it is unlikely that cells would rely on the PPP for ATP production.

Our claim about the glycolytic state of the oligomycin-treated cells is highly supported by the changes in lactate production or glucose consumption rates, both of which would be expected to increase if cells upregulate glycolysis in response to oligomycin treatment. This is something that is very well characterized during measures of the extracellular acidification rate (ECAR), which is often used as a measure of glycolytic activity.

Finally, the authors should define all abbreviations including HD, iDC, OS, PFS, and many more.

Response to Reviewer 4: We apologize for any terms not defined in the text. We will add abbreviations and follow journal policy.

REVIEWER COMMENTS

Reviewer #1 (Remarks to the Author):

The authors have performed additional experiments and revised sections of the manuscript to address the concerns I raised. While some of these concerns have now been sufficiently addressed, others require some additional attention.

Assessment of the functional effects of AMPK inhibition are informative, but are not conclusive. The authors target AMPK with the aim to assess the role of mitochondrial metabolism in DC activation. However, AMPK signaling affects a large number of other cellular processes in addition to mitochondrial biology. So the statement 'selective p-AMPK inhibitor Dorsomorphin to further analyze the consequences of mitochondrial metabolism inhibition on expression of DC immune markers' and 'Collectively, these results suggest that inhibition of mitochondrial metabolism results in impaired expression of surface DC immune phenotype' is not correct. To be able to draw this point, the authors should at least show that AMPK inhibition does affect mitochondrial metabolism (and glycolysis as well) of these cells.

And a general comment to most newly added sections is that the interpretation of these data somewhat remain superficial as there is little comment on whether the data (correlations) make biological sense, or suffer from overinterpretation. For instance, the correlation analyses of the luminex data with the DC metabolic parameters shows inverse correlations with anti-inflammatory mediators (IDO, LAG3 etc). The authors then state that 'metabolic state impacts the immune-related protein secretion profile of DCs' which is not only too strong of a statement (it's only a correlation) but also fails to inform on whether the direction of correlation makes biological sense. If highly glycolytic state of DCs seen in cancer patients which often have an immune suppressed phenotype, these findings seem counterintuitive. Yet, this is not touched upon.

Exactly the same can be said for the correlations between monocyte and DC metabolism, in which the only significant correlations are inverse correlations between monocyte and DC FAO and Gln metabolism. Yet how the authors interpret this finding is unclear.

My main point being that adding more correlations only helps to improve the quality of the

manuscript if it come along with a proper discussion of what the biological implications of these findings.

Please change the heading of the final paragraph of the results section. It is not shown that 'Metabolic changes in monocyte/myeloid circulating populations underlie immune differences between HD and melanoma patients'. There is an association - causation is not shown

And a final poin is that many comparisons and correlations are not significant and show often only trends, This should be clear from the text by using the right wording: ' trend towards' and 'non-significant change' etc. This currently not always the case

Reviewer #2 (Remarks to the Author):

Adamik and colleagues have taken into account most of the comments raised and I am satisfied with their responses and adaptations in the manuscript, although the indications of changes in the text was not always correct.

As a minor suggestion, I propose to carefully read the newly inserted text and adapt where necessary to make it more accurate.

Reviewer's Comments:

Reviewer #1 (Remarks to the Author)

The authors have performed additional experiments and revised sections of the manuscript to address the concerns I raised. While some of these concerns have now been sufficiently addressed, others require some additional attention.

Response: We appreciate the acknowledgement from the reviewer.

Assessment of the functional effects of AMPK inhibition are informative, but are not conclusive. The authors target AMPK with the aim to assess the role of mitochondrial metabolism in DC activation. However, AMPK signaling affects a large number of other cellular processes in addition to mitochondrial biology. So the statement 'selective p-AMPK inhibitor Dorsomorphin to further analyze the consequences of mitochondrial metabolism inhibition on expression of DC immune markers' and 'Collectively, these results suggest that inhibition of mitochondrial metabolism results in impaired expression of surface DC immune phenotype' is not correct. To be able to draw this point, the authors should at least show that AMPK inhibition does affect mitochondrial metabolism (and glycolysis as well) of these cells.

*Response: We understand the critique of the reviewer. To address this, we have now added additional data from p-AMPK inhibition in DC in the form of graph panels to Suppl. Figure 4A (pasted below) in which we also reduce mitochondrial mass, glucose uptake and lactate secretion with p-AMPK inhibition. We have also modified the text as suggested by the reviewer, in which we rephrase our conclusions. The new text is in **bold** below:*

"Because p-AMPK is a well-established positive regulator of mitochondrial health⁴⁵ and metabolism⁴⁶⁻⁴⁸, we employed Dorsomorphin to further analyze the consequences of p-AMPK inhibition on both immune and metabolic phenotype of mDC. Increased concentrations of Dorsomorphin inhibited p-AMPK phosphorylation, and resulted in reduced expression of several DC immune markers including HLA-DR, CD86, PD-L1 and CD206 in HD mDC (Supp Figure 4A). p-AMPK inhibition also exhibited small but significant decrease in mitochondrial mass along with increase in glucose and decrease in lactate levels in media without impact on mDC viability (Supp Figure 4A)."

And

"While we and others have previously shown that p-AMPK associates with mitochondrial metabolism in maturing DC⁴⁸⁻⁵⁰, we further showed that inhibition of p-AMPK resulted in impaired expression of surface immune phenotype and reduced mitochondrial mass of mDC."

And

"Because we also observed reduced glucose uptake along with lower extracellular lactate levels we speculate that blockade of p-AMPK has broad impact on mDC"

metabolism. Additional studies are needed to precisely link and separate the effects of p-AMPK blockade on mitochondrial respiration and glycolysis in mDC.”

And a general comment to most newly added sections is that the interpretation of these data somewhat remain superficial as there is little comment on whether the data (correlations) make biological sense, or suffer from overinterpretation. For instance, the correlation analyses of the luminex data with the DC metabolic parameters shows inverse correlations with anti-inflammatory mediators (IDO, LAG3 etc). The authors then state that 'metabolic state impacts the immune-related protein secretion profile of DCs' which is not only too strong of a statement (it's only a correlation) but also fails to inform on whether the direction of correlation make biological sense. If highly glycolytic state of DCs seems in

cancer patients which often have an immune suppressed phenotype, these findings seem counterintuitive. Yet, this is not touched upon.

Response: We understand the critique of the reviewer. To address this, we have edited the text and more deeply discuss the implications of our findings in a clearer way.

Specifically, we have added the following to the Discussion section regarding the Luminex data:

“Our protein secretion analysis showed several correlations with DC metabolic parameters. LAG3 is an immunosuppressive checkpoint molecule known to block T cell activation, however as a soluble ligand, LAG3 was shown to bind to MHC class II, and promote maturation and CD8-T cell antigen presentation by DC^{71,72}. Consistent with these reports, we hypothesize that soluble LAG3 may be a cell surface indicating optimal maturation of patient-derived mDC, as its level was inversely correlated with glycolytic mDC which exhibited impaired differentiation. A group of immune-suppressive molecules CTLA4, PD-1, IDO, BTLA implicated in inhibition of CD4-T cell proliferation by DCs⁷³ were decreased in mDC cultures that exhibited higher glucose dependence, implying that lower glucose dependence is an immunosuppressive profile of DC vaccines. The functional role of glucose as well as other energy sources including fats and amino acids in immune phenotype and protein secretion requires further dissection. IP-10 has been implicated in regulatory DC-mediated recruitment and inhibition of Th1 cell proliferation⁷⁴ and we observed its positive correlation with maximal oxygen consumption rate along with inverse correlation with TNFSF receptor family members APRIL and TNFB as well as IL-12 family members IL-23 and IL-27, known to influence B cell activation and Th1/2 cell responses^{75,76}. These observations from DC cultures are hypothesis generating, and the functional link between DC metabolic state and protein secretion profiles requires further experimental testing.

Exactly the same can be said for the correlations between monocyte and DC metabolism, in which the only significant correlations are inverse correlations between monocyte and DC FAO and Gln metabolism. Yet how the authors interpret this finding is unclear.

My main point being that adding more correlations only helps to improve the quality of the manuscript if it come along with a proper discussion of what the biological implications of these findings.

Response: We understand the critique of the reviewer. To address this, we have now edited the text to more deeply discuss the correlations and their meaning in our view. As suggested, we have specifically edited and added to the results of circulating monocytes and DC.

Please change the heading of the final paragraph of the results section. It is not shown that 'Metabolic changes in monocyte/myeloid circulating populations underlie immune

differences between HD and melanoma patients'. There is an association - causation is not shown

*Response: To address this, we have changed the final Results paragraph heading to **"Monocyte/myeloid circulating populations exhibit metabolic changes and immune differences between HD and melanoma patients"**.*

And a final point is that many comparisons and correlations are not significant and show often only trends, This should be clear from the text by using the right wording: 'trend towards' and 'non-significant change' etc. This currently not always the case

Response: We acknowledge the critique of the reviewer. To address this, we have now reviewed the figures, legends and text for "trends" vs. significant changes. The primary way we conveyed this information is in the figures with p values which are in each section of each figure. If the journal would prefer asterisks added to the figures by policy, we can add them.

Reviewer #2 (Remarks to the Author)

Adamik and colleagues have taken into account most of the comments raised and I am satisfied with their responses and adaptations in the manuscript, although the indications of changes in the text was not always correct.

As a minor suggestion, I propose to carefully read the newly inserted text and adapt where necessary to make it more accurate.

Response: We appreciate the acknowledgement of the reviewer. We have further reviewed the manuscript as suggested by both reviewers to better flow from original to edited text.

REVIEWERS' COMMENTS

Reviewer #1 (Remarks to the Author):

In response to my additional queries the authors have added some new results and discussion points, that have helped to further improve the quality of the study. I believe it now is of sufficient quality that it meets the standards of Nat Comm.